# Fluorescent protein and peptide tags alter condensate formation and dynamics in vivo and in vitro

Kerstin Dörner [1], Michelle Jennifer Gut [1], Daan Overwijn [1], Fan Cao [2], Matej Siketanc[1], Stephanie Heinrich[1], Nicole Beuret[1], Justin Meyer[1], Timothy Sharpe [1], Kresten Lindorff-Larsen [2] & Maria Hondele [1]✉

## Abstract

Fluorescent proteins and peptide tags are essential tools in cellular biology, but can alter the biochemical and physiological behavior of target proteins. Biomolecular condensates, which have emerged as key elements of cellular organization, are suggested to provide robustness to cells, yet they can also respond sensitively to small changes in environmental conditions including tags. Here, we investigate the effects of over twenty widely used tags on condensate formation in vitro, in cells, in various model organisms and by computational modelling. We find that tagging strongly influences condensation for some proteins, while others remain unaffected. Effects vary, with some tags enhancing and others decreasing condensation, with the outcome depending on the protein being tagged. Coarse-grained simulations suggest that the charge of the fluorescent protein tags is a critical factor modulating condensation behavior. Together, our results underscore the need to tag with caution and highlight the importance of careful experimental design and interpretation, especially in condensate studies, but also suggest that fluorescent protein tags could serve as a tool to modulate condensate properties.

**Keywords** Biomolecular Condensates; Phase Separation; Tags; Fluorescent Protein Tags
**Subject Category** Methods & Resources

## Introduction

Genetically encoded protein and peptide tags have become essential tools in studying protein localization, interaction partners and function, providing an indispensable toolkit for modern cell biology. However, they also pose risks. Tagging introduces steric hindrance and altered chemical and surface properties, which can perturb a protein's physiological behavior, and affect its interactions, folding, turnover, enzymatic activity, and localization (Weller, 2016). A study comparing the localization of roughly 500 human proteins using either immunostaining or fluorescent protein (FP) tagging revealed that approximately 20% of the proteins exhibited different localization patterns between the two methods (Stadler et al, 2013). Also, fixation and permeabilization for immunostaining can affect the perceived distribution of proteins (Irgen-Gioro et al, 2022).

FP tags are widely used in the study of biomolecular condensates such as nucleoli, nuclear speckles (NSs), P-bodies (PBs) or stress granules (SGs). Condensate formation often relies on transient, relatively weak, multivalent interactions that create dynamic viscoelastic structures (Alberti and Hyman, 2021; Mittag and Pappu, 2022; Putnam et al, 2023; Shin and Brangwynne, 2017). The impact of tags on condensates remains somewhat anecdotal and insufficiently documented, with some recent studies of specific proteins highlighting that different tags can considerably alter condensation or aggregation behavior, and introduce artifacts (Dao et al, 2024; Uebel and Phillips, 2019; Ziaunys et al, 2024). For example, tagging huntingtin Httex1(Q25) with RFP results in aberrant condensation (Pandey et al, 2024). mEGFP, FusionRed, mNeonGreen or Halo increased condensation of the viral protein µNS (Barkley et al, 2024), and AID-sfGFP or mEGFP increase and decrease condensation of HP1α, respectively (Zhou et al, 2025). Quantitative in vitro measurements showed differences in the salt-dependency of condensation between hnRNPA1 and His-SUMO-hnRNPA1 (Martin et al, 2021), and a series of genetically encoded phase separation sensors gave different responses depending on which FP was used (Avecilla et al, 2025). Tagging the yeast P-body component Dhh1 with different FPs alters the biophysical properties of Dhh1 condensates in vitro and affects the number of PBs in glucose-starved yeast cells (Fatti et al, 2025). However, a comprehensive comparative analysis of how different commonly used tags affect condensation is lacking.

We therefore systematically tested the influence of over 20 FP, ligand binding and peptide tags on prominent proteins of interest (POI) from human, *Saccharomyces cerevisiae*, and *Escherichia coli*,

[1]Biozentrum, University of Basel, Basel, Switzerland. [2]Structural Biology and NMR Laboratory & the Linderstrøm-Lang Centre for Protein Science, Department of Biology, University of Copenhagen, Copenhagen, Denmark. ✉E-mail: maria.hondele@unibas.ch

using a combination of experiments in cells, in vitro and computational simulations. All proteins tested have been shown to form condensates in vitro (Bearss et al, 2021; Currie et al, 2023; Guillén-Boixet et al, 2020; Heinrich et al, 2024; Hondele et al, 2019; Riback et al, 2017; Sanders et al, 2020; Yang et al, 2020; Yoo et al, 2022). Initially, we examined 16 genetically encoded tags fused to DDX3X in HeLa cells, revealing changes in DDX3X SG appearance that correlated with altered condensation of DDX3X-FP in vitro. Further analysis of human condensate marker proteins showed that while EDC3 (PB) was also affected by FP tagging, G3BP1 (SG) and SRRM2 (NS) were not, indicating that the 'tag effect' may be very POI-dependent. FP tagging influenced the condensation of *S. cerevisiae* Pab1 both in vivo and in vitro, as well as the condensation of recombinant *E. coli* CsdA and human FUS in vitro. Our experiments demonstrate that mCherry consistently reduces condensate formation, while EGFP or EYFP tend to induce larger and potentially more aggregated structures. We also found that common purification tags can alter RNA dependency in condensate formation. Molecular simulations suggest that charge effects can explain some, but not all of the tag effects. This leads us to conclude that for in vitro experiments with recombinant proteins, FP tagging should be avoided if possible. However, tagging remains an invaluable tool in cell biology, including the study of condensates. It is essential to carefully evaluate and validate that the tagged protein closely replicates the behavior of its native counterpart to ensure that it does not interfere with the biological processes being studied.

## Results

### Fluorescent protein, peptide or ligand-binding tags alter DDX3X stress granule appearance

To systematically investigate how the appearance of condensate marker proteins is affected by various FP and peptide tags, we first focused on DDX3X, a DEAD-box ATPase and key SG marker (Ryan and Schröder, 2022). SGs are cytoplasmic condensates (0.1–2 µm) that form in response to stress inducers such as sodium arsenite or heat, and are enriched in mRNAs and RNA-binding proteins, including stalled translation pre-initiation complexes (Buchan and Parker, 2009; Riggs et al, 2020).

As a first step, we screened the effect of 16 different tags C-terminally fused to DDX3X (Table EV1; Appendix Figs. S1 and S2): nine commonly used FPs, five peptide tags, and two ligand-binding tags (Fig. 1A). Initially, we transiently expressed these constructs in HeLa K cells, induced SG formation using sodium arsenite and fixed the cells 24 h post transfection. As a reference, we included an antibody staining against endogenous DDX3X. Compared to the antibody staining, cells expressing tagged DDX3X constructs had considerably altered SG characteristics (Fig. 1A,B). We quantified these changes by measuring the number of SG foci per cell, the average area of individual foci, and an 'intensity enrichment coefficient' (IEC), calculated as the median intensity of each focus divided by the median intensity of the entire cell. The fusion of different FP tags induced strikingly different SG morphologies. ECFP, EGFP, EYFP, and mNeonGreen resulted in larger SGs, and for all but ECFP also in more SG per cell. For ECFP and EYFP the granules became non-spherical, which could suggest

protein aggregation or shape restriction by other cellular structures. mCherry and mScarlet-I reduced SG numbers while maintaining similar SG size, and mRuby3 showed SG numbers comparable to endogenous DDX3X but with slightly enlarged area. All FPs, especially most of those from the red wavelength spectrum, substantially decreased the IEC compared to AB-stained endogenous DDX3X, indicating impaired recruitment of the FP-tagged DDX3X to SG (Fig. 1B). The divergent behavior of red and green fluorescent proteins likely reflects differences in their evolutionary origins rather than their emission wavelength, as many green variants are derived from avGFP and thus likely share similar biophysical properties (Appendix Fig. S2). In summary, tagging with any FP strongly altered the appearance of DDX3X SGs, with mEGFP likely providing the least disruptive tag option.

Peptide tags overall had a more uniform and less variable influence, typically increasing the number of DDX3X SG foci per cell (Fig. 1A,B). In a subset of cells, HA, FLAG, and ALFA led to the formation of a few larger and irregularly shaped foci, hinting at protein aggregation. Myc and V5 behaved quite similarly to the endogenous protein across all parameters, indicating they could be good tag choices.

The ligand-binding tags Halo and SNAP were visualized using either anti-tag antibodies or tag-binding dyes (Fig. 1C). DDX3X-Halo/SNAP SG foci were smaller and more irregular compared to endogenous DDX3X (Fig. 1C,D), which could suggest aggregation. Notably, visualization with dyes significantly reduced both SG numbers and IEC values, indicating impaired recruitment of tagged DDX3X to the granules (Fig. 1D; Appendix Fig. S3A). In conclusion, SNAP and Halo tags appear suboptimal for studying DDX3X in cells.

Leveraging the variability in expression levels across cells in transient transfections, we next assessed how SG characteristics correlate with DDX3X expression levels (Fig. EV1A,B). While endogenous DDX3X displayed a relatively uniform distribution of SG number and area, transiently expressed tagged DDX3X showed greater variability of these parameters across expression levels. Surprisingly, for most tags, we observed only a weak correlation between expression levels and SG number or area, which could suggest that tagged DDX3X is primarily recruited to pre-existing membraneless organelles (MLOs), rather than inducing de novo condensate formation as a result of overexpression.

### Titrating expression levels reduces variability in DDX3X stress granule appearance

Considering the phenotypic variability observed in transiently transfected cells, we generated stable HeLa cell lines expressing DDX3X tagged with a selection of tags from a doxycycline-inducible promoter. We titrated expression levels to match endogenous DDX3X at a 1:1 ratio, with low variability between individual cells (Fig. 1E,F; Appendix Fig. S3B–E). Furthermore, to rule out fixation-induced artifacts (Irgen-Gioro et al, 2022) that might differentially affect fluorescent proteins and thereby alter MLO appearance, we conducted the experiments in both fixed and living cells (Fig. EV1C,D). The results were largely consistent with those observed in transient transfections, especially for DDX3X-mCherry which still showed a drastic reduction of SG number, area and IEC. For the other FP and peptide tags the effects were considerably milder (Figs. 1E,F and EV1C,D), suggesting that even

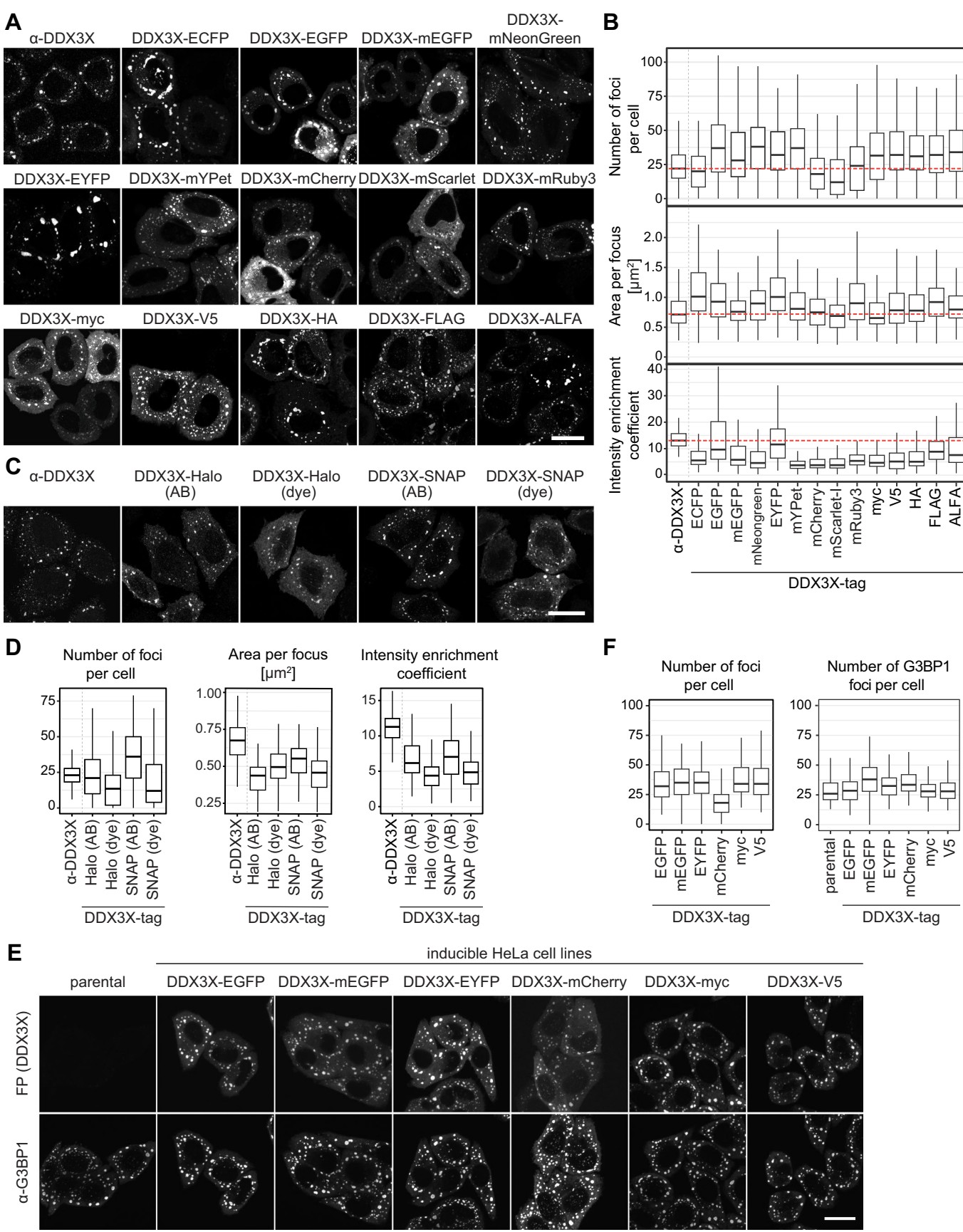

**Figure 1.  Expression of DDX3X with various tags influences stress granule formation in human cells.**

Transient or stable inducible expression of DDX3X constructs for 24 h in HeLa cells. Cells were stressed with 500 μM sodium arsenite for 30 min before PFA fixation. For each experiment, three independent replicates were performed ($N = 3$). Quantification of DDX3X foci includes number of foci per cell, area per focus and intensity enrichment coefficient (IEC, median intensity per focus/ median intensity of the cell). Box plots show the median (center line), 25th–75th percentiles (bounds of box), and whiskers=1.5× IQR.(A) Transient expression of DDX3X fused with FP and peptide tags in HeLa K cells. Untransfected cells were immunostained for DDX3X. Peptide tags were visualized by immunostaining with the respective antibodies. Scale bar: 20 μm. (B) Quantification of (A); $N = 3$, $n \geq 135$ cells. (C) Transient expression of DDX3X fused with Halo and SNAP tag in HeLa K cells. Tags were either visualized by incubation with respective dyes (dye) or by immunostaining (AB). Scale bar: 20 μm. (D) Quantification of (C); $N = 3$, $n \geq 90$ cells. (E) Stable inducible HeLa cell lines were induced with doxycycline for 24 h to express the respective DDX3X-tag constructs at endogenous level. Cells were stressed with 500 μM sodium arsenite for 30 min before fixation. For visualization of SGs cells were additionally immunostained for G3BP1. Representative maximum Z-projections are shown. Scale bar: 20 μm. (F) Quantification of (E) $N = 3$, $n \geq 65$ cells. Source data are available online for this figure.

low levels of transient expression may still represent significant overexpression.

To test whether endogenous DDX3X is affected by exogenous expression of FP-tagged DDX3X, we stained the stable cell lines with a DDX3X antibody. The resulting DDX3X immunofluorescence signal was very similar to that of the DDX3X-FP channel (Fig. EV1E,F), suggesting that overall endogenous DDX3X levels and localization are not substantially different from FP-tagged DDX3X. We then immunostained for another SG marker, G3BP1, which also showed minimal differences in SG appearance between parental and DDX3X-FP cells (Fig. 1E,F; Appendix Fig. S3F). In both co-stainings, we observed a marked reduction in SG-enrichment for DDX3X-mCherry compared to the antibody signal (Appendix Fig. S3D,F). This suggests that while recruitment of DDX3X-mCherry is reduced, localization of endogenous SG proteins, such as G3BP1 and endogenous DDX3X, remains unaffected.

### General considerations for studying condensation in vitro

We wondered whether the effects observed in cells might originate from inherent differences in the condensate-forming ability of DDX3X fusion constructs. To investigate this, we purified DDX3X untagged and tagged with various FPs (Fig. EV2A). Condensates formed from untagged proteins can be visualized by brightfield or quantitative phase microscopy (McCall et al, 2025). However, widefield or confocal microscopy of untagged proteins cannot analyze multi-component condensates, or critical biophysical parameters such as the partition coefficient (PC) or component turnover dynamics. Furthermore, untagged condensates present difficulties in maintaining proper focus during automated microscopy workflows. Therefore, we introduced a spike-in of 1% POI chemically labeled with ATTO NHS-ester dyes. Given that we observed no differences in condensation behavior between the untagged and untagged+ATTO samples for DDX3X (Fig. EV2B), we designated untagged+ATTO as the reference point for all subsequent in vitro experiments. For calculations of the PC and FRAP analysis, we assumed that the untagged+ATTO samples behave like untagged samples.

It is important to keep in mind that the brightness of certain FPs is pH-sensitive. For example, at lower pH (pH 6–7), EGFP and EYFP appear much dimmer compared to higher pH (pH 7–8) (Campbell and Choy, 2001). In this manuscript, we display images with identical exposure settings for all panels of one FP, thus likely underestimating the intensity of EGFP and EYFP at high pH compared to low pH. For FUS (Fig. 5), intensities were so different that we adjusted exposure levels individually for each image. As

previously reported (Hyman et al, 2014), condensates undergo growth and fusion over time (Fig. EV2C). Therefore, within an experiment, it is critical to precisely control and standardize the time between condensate formation and image acquisition for each individual sample to ensure consistency in experimental observations.

### Fluorescent protein tags modify the characteristics of in vitro DDX3X condensates

We examined the condensation behavior of DDX3X with polyuridylic acid (poly(U)) across varying pH and salt conditions (Fig. 2A–D). Untagged+ATTO DDX3X formed condensates at pH 6.2 and low salt, but tended to aggregate at higher pH (Fig. 2A,B) and dissolved at higher salt (Figs. 2C,D and EV2D,E). DDX3X tagged with the highly related EGFP or EYFP (Appendix Fig. S1 and S2) showed similar behavior but typically formed larger condensates with higher PC and increased salt resistance. DDX3X tagged with mEGFP formed large, round condensates across all pH conditions, showing less sensitivity to pH and salt compared to untagged+ATTO DDX3X and DDX3X-EGFP. DDX3X-mCherry produced the largest and 'roundest' condensates at low salt and pH, suggesting enhanced fusion, but these shrank and disappeared at higher pH and salt levels (Fig. 2C,D). These findings suggest that the DDX3X-FP SG phenotypes observed in cells, where the cytoplasm has a pH of around 7.2, may be at least partly driven by the altered condensation behavior of DDX3X itself.

Since some FPs are known to dimerize (Campbell and Choy, 2001), we conducted analytical ultracentrifugation (AUC) in buffers similar to those used for DDX3X condensation assays (150 mM NaCl, pH 7.4 and 6.2) for the isolated FP proteins (Fig. EV2F,G). EGFP and EYFP showed partial dimerization as previously reported (Campbell and Choy, 2001), while mEGFP and mCherry remained monomeric. This dimerization likely contributes to the pronounced effects of EGFP and EYFP on DDX3X condensation, but is probably just one of several factors (see below) (Table EV1; Appendix Fig. S1).

To qualitatively evaluate the material properties of DDX3X-mCherry versus untagged+ATTO, we first conducted a time-course analysis of condensate formation (Fig. EV2C). Both proteins underwent condensate fusion, but DDX3X-mCherry formed larger condensates while smaller ones completely disappeared, whereas untagged+ATTO DDX3X retained smaller, more solid-like condensates. To investigate material properties in more detail, we analyzed condensate turnover by fluorescence recovery after photobleaching (FRAP) (Fig. 2E,F). Recovery kinetics varied substantially between the proteins:

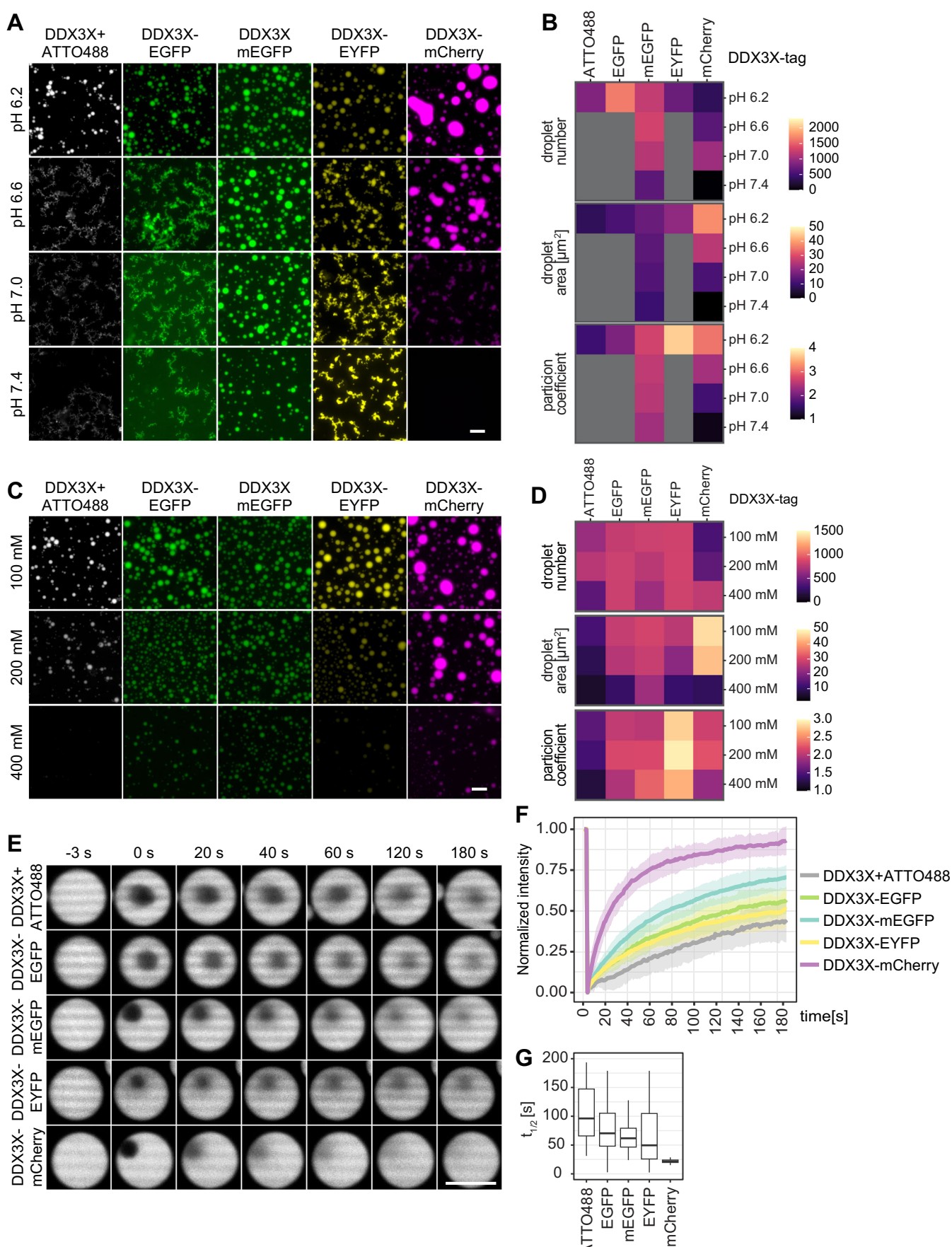

**Figure 2. Fluorescent protein tags influence DDX3X condensate formation and turnover in vitro.**

DDX3X in vitro condensation assays with untagged+1% ATTO488 spike-in, or FP-tagged proteins were incubated at 25 °C for 30 min before imaging. Quantification includes number of condensates per 0.1 mm$^2$, droplet area and mean PC. Box plots show the median (center line), 25th–75th percentiles (bounds of box), and whiskers = 1.5× IQR. (A) 5 µM DDX3X in 25 mM sodium phosphate buffer at the indicated pHs, 150 mM NaCl, 2 mM MgCl$_2$, 0.5 mg/mL BSA, 0.05 mg/mL poly(U), 0.5 mM ATP, 0.5 mM DTT. Incubated at 25 °C for 30 min before imaging. Scale bar: 20 µm. (B) Quantification of (A): images displaying aggregates were not quantified and are shown in grey. Four well positions were imaged in three independent replicates. (C) 5 µM DDX3X in 25 mM sodium phosphate buffer at pH 6.2, at different salt concentrations (100, 200 or 400 mM NaCl), 2 mM MgCl$_2$, 0.5 mg/mL BSA, 0.05 mg/mL poly(U), 0.5 mM ATP, 0.5 mM DTT. Scale bar: 20 µm. (D) Quantification of (C): four well positions were imaged in three independent replicates. (E) FRAP of in vitro DDX3X condensates of similar size. 5 µM DDX3X in 25 mM sodium phosphate buffer at pH 6.2, 100 mM NaCl, 2 mM MgCl$_2$, 0.5 mg/mL BSA, 0.05 mg/mL poly(U), 0.5 mM ATP, 0.5 mM DTT. Condensates were matured for 1 h at 25 °C before FRAP. Scale bar: 5 µm. Fluorescence recovery curve of DDX3X droplets (F) and time till half recovery of DDX3X droplets (G) were analyzed. $N = 3$, $n \geq 23$ condensates. Source data are available online for this figure.

untagged+ATTO DDX3X had a relatively slow recovery ($t_{1/2} \sim 100$ s), while FP-tagged variants recovered faster: EGFP, mEGFP, and EYFP with $t_{1/2}$ between 50-70 s, and mCherry with a $t_{1/2}$ of ~25 s (Fig. 2G). FP tags also influenced the mobile fraction of DDX3X, with DDX3X-mCherry fully recovering within 180 s, while other constructs reached lower plateau levels, suggesting reduced mobility and potential aggregation (Fig. EV2H). These findings align with the observation that untagged+ATTO, EGFP, and EYFP-tagged DDX3X seem aggregated at higher pH, and indicate that aggregated states may be present at pH 6.2 already, where FRAP was conducted.

In conclusion, FP tags have a substantial impact on DDX3X condensate appearance. At physiological pH, DDX3X-mCherry condensates do not form, both in vitro and in cells. In contrast, EYFP and EGFP tags seem more aggregated and have slower in vitro FRAP recovery compared to DDX3X-mCherry condensates, which might explain the enlarged SGs observed in cells upon transient overexpression. This highlights the possibility of screening FP tags in vitro before conducting cellular experiments to select those that minimally affect protein behavior.

## mCherry-EDC3 strongly reduces P-body numbers and recruitment of DCP1 and DDX6

We wondered whether the effects of FP tagging observed with DDX3X extend to other condensation-prone proteins. To explore this, we used a smaller set of four FP and two peptide tags and fused them to various nuclear and cytoplasmic proteins involved in condensate formation.

EDC3 is a scaffold protein of P-bodies (PBs), cytoplasmic granules containing factors involved in mRNA decay and turnover that typically enlarge in stress conditions (Buchan et al, 2008; Riggs et al, 2020). EDC3 fusion proteins were transiently expressed for 24 h and cells were stressed with sodium arsenite before fixation (Fig. 3A). mCherry-tagging resulted in a marked reduction of PB numbers (Fig. 3B), while other FP and peptide tags had minimal impact on PB number, area or IEC (Figs. 3B and EV3A). Similar to DDX3X, the 'tag effects' observed with EDC3 did not correlate with transient expression levels (Fig. EV3B,C).

Stable cell lines expressing tagged EDC3 at endogenous levels confirmed that mCherry considerably reduced PB foci numbers (Fig. 3C,D; Appendix Fig. S4B–D), and surprisingly demonstrated that mCherry-EDC3 expression also drastically reduced the number of co-stained DCP1 or DDX6 foci, while other FPs led to a slight increase (Fig. 3C,D and EV3E; Appendix Fig. S4C,D). Again, similar results were obtained in both in fixed and in living

cell lines (Fig. EV3C,D). This suggests that EDC3 tagging affects not only its own targeting to PBs but also the recruitment of other proteins such as DCP1 or DDX6.

## Tagging G3BP1, DDX6 or SRRM2 has minimal influence on stress granules, P-bodies and nuclear speckles, respectively

Tagging marker proteins does not always have a strong impact on condensate appearance, as illustrated by the following three examples: G3BP1 (SG), DDX6 (PB) and SRRM2 (NS). Stress granule formation upon sodium arsenite treatment requires condensation of G3BP1 (Guillén-Boixet et al, 2020; Sanders et al, 2020; Yang et al, 2020). In our experiments, C-terminal fusion of G3BP1 to various FP and peptide tags did not notably affect the number, area or IEC of G3BP1 SG foci (Fig. 3E,F; Appendix Fig. S4E–G). Similarly, C-terminal tagging of DDX6 did not alter PB appearance (Fig. EV3F,G).

SRRM2 is a scaffold protein of nuclear speckles (NSs), a nuclear condensate involved in splicing and storage of splicing factors (Dörner and Hondele, 2024; Ilik et al, 2020). Analysis of SRRM2-FP foci from transient transfections revealed that SRRM2 expression levels had a greater impact on NS characteristics than the tags themselves (Fig. 3G,H; Appendix Fig. S5A–C). Higher expression levels consistently resulted in larger, fewer NSs, displaying distinct SON and SRRM2 sub-phases, consistent with the literature (Zhang et al, 2024).

In summary, while some condensate marker proteins are highly sensitive to tagging, others are minimally affected, even if they localize to the same condensate (e.g. DDX3X versus G3BP1 or EDC3 versus DDX6). The impact of tagging varies across POIs, and tagging can also influence other proteins recruited to the same organelle (e.g. also the DCP1 and DDX6 signal is altered when EDC3 is tagged with mCherry). Furthermore, the results described above show that the effect of tagging can also depend on the expression levels.

## FP tagging of Pab1 alters SG formation in yeast, with mCherry strongly reducing granule formation

We wanted to investigate whether perturbations resulting from the expression of fusion proteins also appear in yeast. Therefore, we tagged *S. cerevisiae* Pab1, the cytoplasmic poly(A) binding protein and a prominent SG marker, with various FPs at the endogenous locus (Fig. 4A). Using live cell imaging, we observed striking differences in SG numbers depending on the FP used: ECFP,

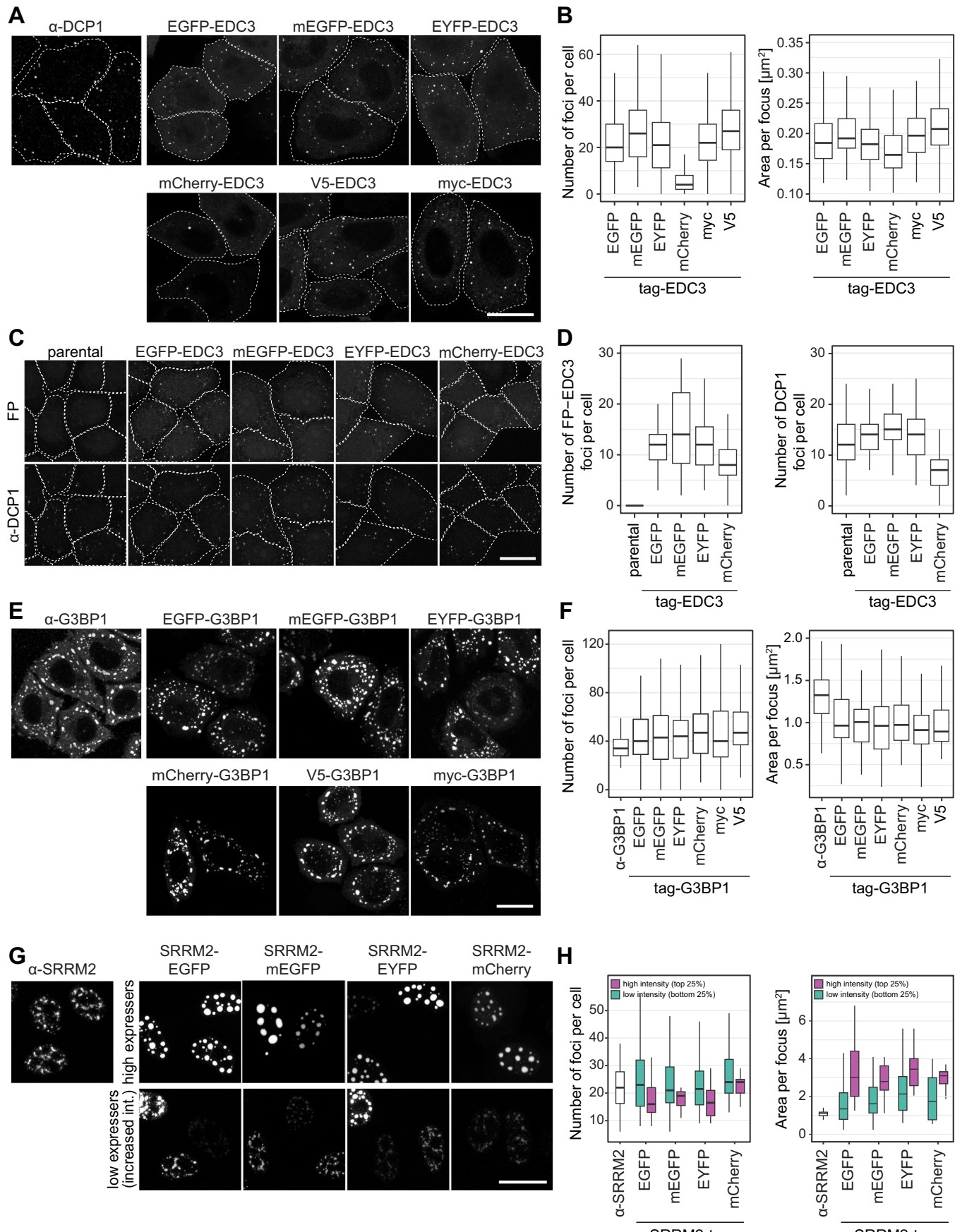

◄

**Figure 3. mCherry tagging reduces EDC3 foci but does not affect G3BP1 or SRRM2 foci formation.**

Quantifications of foci include number of foci per cell and area per focus as indicated. Box plots show the median (center line), 25th–75th percentiles (bounds of box), and whiskers = 1.5× IQR. (A) HeLa K cells were transiently transfected with EDC plasmids for 24 h, and stressed with 500 μM sodium arsenite for 30 min before fixation. V5-/myc-EDC3 were visualized by immunostaining with the respective antibodies. Untransfected cells were immunostained for DCP1 to visualize PB. Scale bar: 20 μm. (B) Quantification of (A); $N = 3$, $n \geq 110$ cells. (C) Stable inducible HeLa cell lines were induced with doxycycline for 24 h to express the respective EDC3-tag constructs at endogenous level. Cells were stressed with 500 μM sodium arsenite for 30 min. For visualization of PB, cells were additionally immunostained for DCP1. Unfortunately, the two EDC3 antibodies we tested only worked with methanol fixation, which is often incompatible with FP fluorescence. Therefore, we decided to use an antibody against the co-localizing PB marker protein DCP1 as a reference (Appendix Fig. S4A). Scale bar: 20 μm. (D) Quantification of (C). $N = 2$, $n \geq 55$ cells. (E) HeLa K cells were transiently transfected with G3BP1 plasmids for 24 h and stressed with 500 μM sodium arsenite for 30 min before fixation. V5-/myc-G3BP1 were visualized by immunostaining with the respective antibodies. Untransfected cells were immunostained for G3BP1 as a control. Scale bar: 20 μm. (F) Quantification of (E); $N = 3$, $n \geq 90$ cells. (G) HeLa K cells were transiently transfected with SRRM2 plasmids for 24 h and stressed with 500 μM sodium arsenite for 30 min before fixation. Untransfected cells were immunostained for SRRM2. Scale bar: 20 μm. (H) Quantification of (G) binned into low and high overexpressing cells; $N = 3$, $n \geq 40$ cells. Source data are available online for this figure.

mTurquoise, and EGFP tags resulted in similar SG counts, EYFP doubled the count, mYPet caused a strong reduction, and mCherry or mScarlet-I nearly eliminated SGs (Figs. 4B and EV4A,B). Similar but not identical trends were observed when comparing two genetic backgrounds, W303 (Fig. 4A,B) and BY (Fig. EV4A,B).

Next, we examined the condensation behavior of recombinant Pab1 in vitro (Figs. 4C and EV4C,D). Consistent with previous findings, Pab1 formed hydrogel-like structures rather than round-ish condensates (Figs. 4C and EV4D) (Riback et al, 2017; Yoo et al, 2022). Surprisingly, tagging substantially altered even these more aggregated structures: EGFP, mEGFP, and EYFP-tagged Pab1 formed clusters similar in abundance to untagged+ATTO Pab1, although they appeared somewhat larger and rounder. In contrast, mCherry tagging drastically reduced Pab1 assemblies (Figs. 4C and EV4D). These results emphasize the need for careful validation of FP tags for both in vivo and in vitro experiments.

## His-TwinStrep affinity tag enhances Nab2 condensate formation

*S. cerevisiae* Nab2, the nuclear poly(A)-binding protein, forms condensates in vitro and likely in cells (Heinrich et al, 2024). FP tags were not tested in this case, but we found that a commonly used purification affinity tag, 6xHIS-TwinStrep (HisTS), can affect the RNA dependency of Nab2 condensate formation. HisTS-Nab2-ymEGFP and Nab2-ymEGFP, after cleavage of the HisTS tag, (Fig. EV4E,F) were tested for condensate formation in the presence of poly(A) or poly(U) RNA, and without RNA (Fig. 4D,E). While cleaved Nab2-ymEGFP formed condensates only with its 'natural' substrate poly(A), HisTS-tagged Nab2-ymEGFP formed condensates in all tested conditions, and produced large assemblies with poly(A). This suggests that the HisTS tag adds nonspecific multivalency to Nab2, thereby altering its RNA dependency.

## FP tags modulate in vitro condensation of *E. coli* CsdA and to a lesser extent human FUS

We extended our in vitro study to two proteins that form condensates even in relatively high salt conditions, *E. coli* CsdA and human FUS. First, we tested CsdA fused with nine FP tags across different pH values (Fig. 5A; Appendix Fig. S6A). Untagged+ATTO CsdA and CsdA fused with mEGFP, EBFP2, mKO2, and mNeonGreen formed condensates of similar size and PC (Fig. 5A,B). In contrast, tagging with mYPet, mTurquoise2, and particularly mScarlet-I, mApple, and mCherry resulted in smaller,

fewer condensates with lower PC, and at pH 8.0, these tags almost completely abolished condensation. A more detailed analysis of a subset of these FP tags revealed that while untagged+ATTO CsdA condensation was stable up to 400 mM salt across all pH values tested, FP-tagged CsdA showed increased sensitivity to salt, particularly at higher pH (Fig. EV5A; Appendix Fig. S6B). Notably, some FP tags, such as mEGFP, mCherry, and mYPet, affected condensate stability more than others like EBFP2 or mKO2. In summary, FP tags tend to reduce CsdA condensation, with the most pronounced effect at high salt and pH.

Next, we examined the in vitro condensation of FUS, an RNA binding protein implicated in amyotrophic lateral sclerosis (ALS) and frontotemporal dementia (FTD) (Kwiatkowski et al, 2009; Vance et al, 2009), and one of the most-studied proteins in condensation research (Fig. 5C,D; Appendix Fig. S7A,B). At lower salt concentrations, FUS tagged with EGFP, mEGFP, or EYFP formed condensates similar to untagged+ATTO FUS. Under high salt conditions (500/1000 mM NaCl), FUS-mEGFP remained comparable to untagged+ATTO FUS, whereas EYFP/EGFP tagged FUS showed increased droplet area and PC, especially at higher pH. Again, the mCherry tag strongly reduced condensation compared to other FUS variants. Overall, although FUS showed some sensitivity to FP tags, it was less affected than CsdA or DDX3X, likely due to its more robust condensation. Based on our analysis, the mEGFP tag might be a good choice for cellular studies of FUS as it most closely mimicked the behavior of untagged+ATTO FUS in vitro.

## Comparison of in vitro condensation results for mEGFP and mCherry

To better visualize the effects of two commonly used FP tags, mEGFP and mCherry, we compared their influence on the in vitro condensation of DDX3X, FUS, and CsdA (for POI domain structures, see Appendix Fig. S8) at 200 mM salt and two pH levels (6.0/6.2 and 7.4/7.5), normalized relative to the untagged+ATTO samples at the lower pH (Figs. 5E and EV5B). mEGFP tagging increased droplet area and PC for DDX3X, yet it had minimal impact on FUS and CsdA condensation. mCherry tagging had a more pronounced and more complex effect. At low pH, mCherry induced large condensates with increased PC for DDX3X, while FUS and CsdA showed only moderate changes. At physiological pH (7.4/7.5), mCherry consistently impaired condensation for all three proteins. This highlights that mEGFP appears to be a relatively neutral tag for the tested POIs, whereas mCherry severely diminishes or even prevents their condensation.

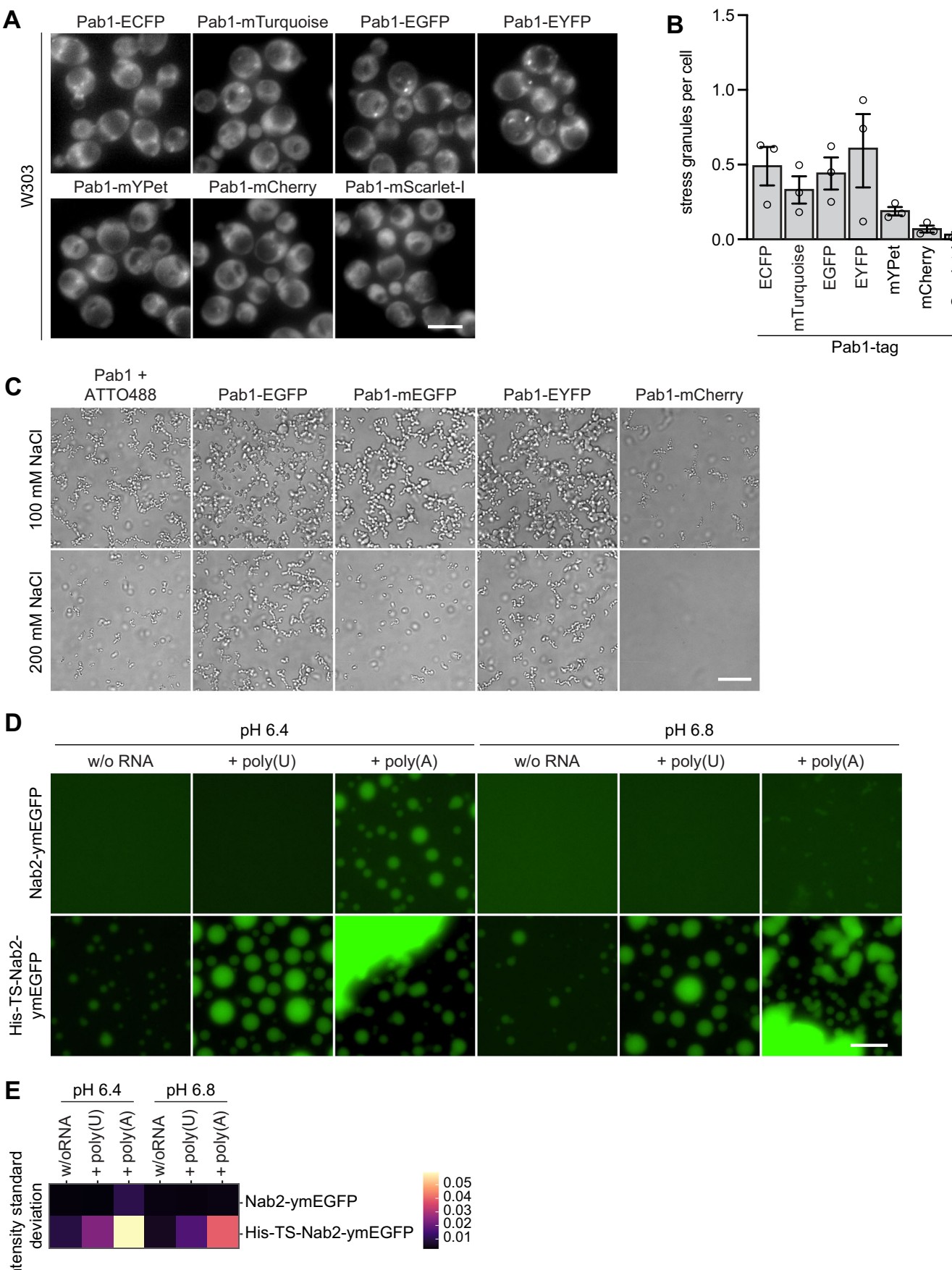

**A** Pab1-ECFP  Pab1-mTurquoise  Pab1-EGFP  Pab1-EYFP
W303
Pab1-mYPet  Pab1-mCherry  Pab1-mScarlet-I

**B**

**C** Pab1 + ATTO488  Pab1-EGFP  Pab1-mEGFP  Pab1-EYFP  Pab1-mCherry
100 mM NaCl
200 mM NaCl

**D**
pH 6.4  pH 6.8
w/o RNA  + poly(U)  + poly(A)  w/o RNA  + poly(U)  + poly(A)
Nab2-ymEGFP
His-TS-Nab2-ymEGFP

**E**
intensity standard deviation
pH 6.4  pH 6.8
w/o RNA  + poly(U)  + poly(A)  w/o RNA  + poly(U)  + poly(A)
Nab2-ymEGFP
His-TS-Nab2-ymEGFP

**Figure 4. Tagging influences condensates of yeast polyA binding proteins in vivo and in vitro.**

(A) *S. cerevisiae* W303 strains expressing tagged Pab1 were cultivated with 3% glycerol for 45 min prior to imaging to induce SG formation. Scale bar: 5 µm. (B) Quantification of SG per cell in (A), mean $+/-$ SEM, $N = 3$, $n \geq 1500$ cells per strain. (C) In vitro condensation assay with 15 µM Pab1 (FP-tagged or untagged + 1% ATTO488-Pab1 spike-in) in 25 mM NaOAc Buffer pH 5.0, at different salt concentrations (100 mM or 200 mM NaCl), 2 mM $MgCl_2$. Incubation at 30 °C for 15 min before imaging. $N = 3$. Scale bar: 20 µm. (D) Comparison of uncleaved Nab2 (His10-TwinStrep(TS)-Nab2-ymEGFP) and cleaved Nab2 (Nab2-ymEGFP). In vitro condensation assay with 20 µM protein in 25 mM sodium phosphate buffer pH 6.4 or 6.8, 50 mM NaCl, 2 mM $MgCl_2$, 0.5 mg/mL BSA with 0.05 mg/mL poly(U) or poly(A) as indicated. Incubated at 20 °C for 20 min before imaging. Scale bar: 20 µm. (E) Quantification of (D): image intensity standard deviation, $N = 3$. Source data are available online for this figure.

## Phase coexistence simulations of fusion proteins

Given these substantial 'FP tag effects', we wanted to gain a better understanding of the underlying biophysical mechanisms. Building on previous studies establishing a molecular grammar for phase separation (Pappu et al, 2023) and molecular simulation methods that relate protein sequence to phase separation propensities (Cao et al, 2024), we performed direct-coexistence simulations on phase-separating proteins or intrinsically disordered regions (IDRs) fused to various FPs using the CALVADOS (Coarse-graining Approach to Liquid–liquid phase separation Via an Automated Data-driven Optimization Scheme) 3 coarse-grained model (Cao et al, 2024). CALVADOS 3 is a coarse-grained (one bead per residue) molecular dynamics simulation model that generates conformational ensembles of intrinsically disordered proteins (IDPs) and multi-domain proteins, from which the phase behavior of these proteins can be inferred. It incorporates a harmonic potential for bonded interactions, an Ashbaugh-Hatch (AH) potential to model van der Waals and hydrophobic forces, a Debye-Hückel (DH) potential for salt-screened electrostatic interactions, and an additional harmonic potential to maintain the structural integrity of folded domains within multi-domain proteins Cao et al, 2024. First, we attached various FPs to either the N- or C-terminus of full-length DDX3X and FUS, proteins which consist of both folded domains and IDRs. The resulting equilibrium density profiles show that FP-tagging influenced protein condensation, albeit to varying degrees (Figs. 6A and EV6A): DDX3X, with its complex amino acid composition and folded domain surface composition, was more strongly affected than FUS, which nicely aligns with our biochemical data (Table EV2; Appendix Figs. S9 and S10). While FP fusions at both the N- and C-termini of FUS generally exhibited similar condensation behavior, DDX3X showed noticeable differences depending on the tagging location. For instance, when ECFP, mEGFP, mCherry, or mScarlet are fused to the C-terminus of DDX3X, the simulations do not show stable condensation. In contrast, fusing the same FPs to the N-terminus results in more robust phase separation. These findings suggest that the distinct phase behavior of DDX3X and FUS upon N- or C-terminal tagging arises from local differences in molecular properties near the termini, such as amino acid composition, charge distribution, and domain architecture. Interestingly, the DDX3X simulations predicted that mRuby3 is less disruptive than other red fluorescent proteins, which was already suggested by the cellular data (Fig. 1A,B) and is further supported by in vitro experiments with CsdA (Fig. EV6B).

Previous work showed that proteins with a net charge close to neutral appear to undergo homotypic phase separation more easily than charged proteins (Bremer et al, 2022; von Bülow et al, 2025; Crabtree et al, 2023; Tesei et al, 2021). We therefore tested to what extent the net charge of the POI and the FP can explain the effects of tagging. We shifted our focus to simpler systems consisting solely of IDRs, specifically the sequences -4D (Bremer et al, 2022) from the prion-like low-complexity domain of human hnRNPA1 (four native Asp residues substituted with Gly/Ser; net charge +12), and the isolated IDR of DDX4 (Brady et al, 2017) (net charge -4), each fused to differently charged GFP variants (GFP$^{-20}$, GFP$^{-15}$, GFP$^{-10}$, GFP$^{-5}$, GFP$^{0}$, GFP$^{+5}$, GFP$^{+10}$, GFP$^{+15}$ and GFP$^{+20}$, superscript numbers indicate net charge). We note that even if GFP$^{0}$ has a formal net charge of zero, it contains both positively and negatively charged residues. In line with previous findings on the recombinant protein (Bremer et al, 2022), untagged -4D undergoes phase separation during the simulations. Fusion with GFP reduces in silico condensation, an effect that is further exacerbated by increasingly positive GFP variants, likely due to electrostatic repulsion of the now highly positively charged fusion proteins (Figs. 6B and EV6C; Appendix Figs. S9–S12). In contrast, attaching negatively charged GFP variants is predicted to significantly promote -4D phase separation, presumably by a combined effect of diminishing the net charge of the fusion protein and favorable interactions between GFP and the disordered region of -4D (Appendix Figs.S11 and 12). Similarly, untagged DDX4 IDR is also predicted to undergo phase separation, in agreement with data for the recombinant protein (Brady et al, 2017). The attachment of most GFP charge variants enhanced condensation, presumably facilitated by additional GFP-IDR interactions (Figs. 6B,C and EV6C,D; Table EV3). As for -4D, we observe that a protein with a net charge close to zero has the strongest driving force for phase separation, but that DDX4 IDR generally showed smaller effects. Interestingly, we find that for -4D the driving force becomes stronger as the net charge is titrated below zero, presumably reflecting additional attractive charge–charge interactions that overcome the global net charge effects. Together, these results demonstrate that net and surface charges of the FP and the POI can dominate the extent of condensation modulation, though other effects such as the number of charged residues also play a role.

## Discussion

Our study demonstrates that tagging a POI, especially with FPs, can considerably alter its condensation properties in vitro and in cells. Phase equilibria depend on interactions both within the condensate and with the surrounding environment, and consequently tags could influence condensate formation by altering interactions among the condensate building molecules, or with other cellular components. Considering that our in vitro and in silico experiments typically reflect the phenotypes observed in cells, it is likely that perturbed homotypic condensation is a key contributing factor. This seems plausible given the densely packed, multivalent

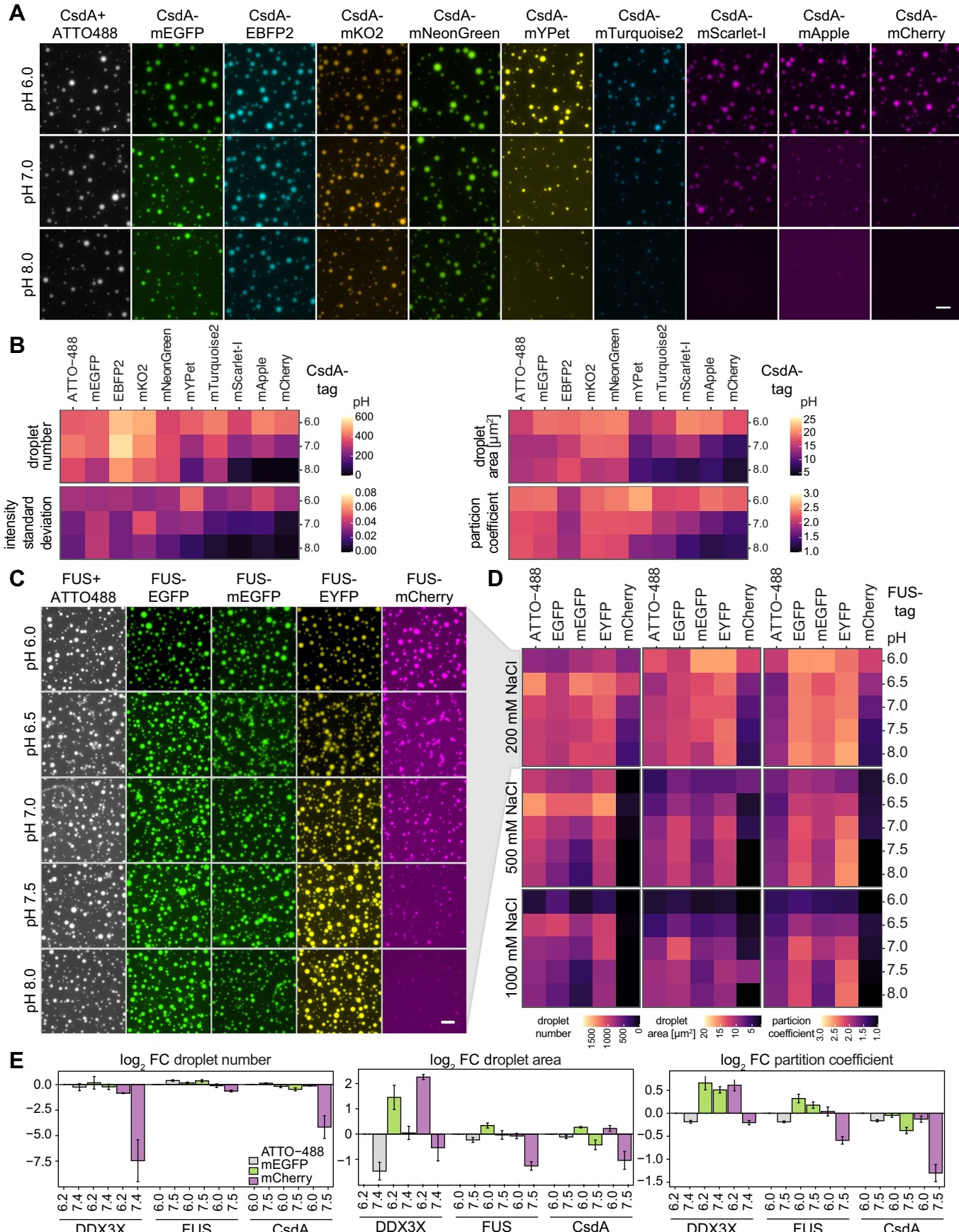

Figure 5.  **Fluorescent protein tags influence the condensation of *E. coli* CsdA and human FUS in vitro.**

(**A**) In vitro condensation assay with 2 µM CsdA (tagged or untagged + 1% ATTO488-CsdA spike-in) in 25 mM sodium phosphate buffer at the indicated pHs, 200 mM NaCl, 2 mM MgCl$_2$, 0.5 mg/mL BSA, 0.05 mg/mL poly(U). Incubated at 25 °C for 1 h before imaging. Scale bar: 20 µm. (**B**) Quantification of (**A**): number of droplets per 0.1 mm$^2$, droplet area, image intensity standard deviation and PC, $N = 3$. (**C**) In vitro condensation assay with 8 µM FUS (tagged or untagged + 2% ATTO488 FUS spike-in) in 25 mM sodium phosphate buffer at the indicated pHs, 200 mM NaCl, 2 mM MgCl$_2$. Condensation was induced by addition of 10 µM His-3C-PreScission protease removing the MBP tag. Incubated at 25 °C for 30 min before imaging. To account for pH-dependent FP intensity differences, we adjusted image brightness individually; corresponding images with consistent settings and DIC are in Fig. EV5D. Scale bar: 20 µm. (**D**) Quantification of (**C**) and (Appendix Fig. S7A): number of droplets per 0.1 mm$^2$, droplet area and PC, $N = 3$. (**E**) Quantification of comparative assessment of DDX3X, FUS, and CsdA in vitro condensation when tagged with mEGFP or mCherry at 200 mM NaCl at pH 6.2/7.4 for DDX3X and pH 6.0/7.5 for FUS and CsdA, representative images are displayed in Fig. EV6B. Log$_2$ fold change (FC) of tagged protein against the respective untagged+ATTO pH 6.2/6.0 sample. Mean +/− SEM, $N = 3$. Source data are available online for this figure.

interaction networks within condensates: tagging their building blocks with large proteins that possess distinct surface properties may easily disrupt these transient interactions. However, 'the tag problem' is not unique to condensates and is a common issue in studies of both individual proteins and multi-protein complexes, including for example the localisation of various proteins (Cui et al, 2016; Stadler et al, 2013; Weill et al, 2019), the polymerisation of actin and tubulin (Abe and Hashimoto, 2005; Westphal et al, 1997) or the functionality of transcription factors (de Folter et al, 2007; Meyer et al, 2007). In fact, condensates may offer a more precise and quantitative framework for analyzing these effects, resulting in a better understanding of how experimental conditions, including tagging, can impact biological processes.

We find that the effects of tagging are complex and context-dependent, varying with the tag, the POI and the surrounding physico-chemical environment. Differences likely arise from alterations in the molecular surface properties of the tagged POI—such as surface charge or hydrophobicity—as well as from the tag's position relative to condensation-mediating residues (see Appendix Fig. S8). Consequently, tagging at the N- versus C-terminus can lead to markedly different outcomes. Obviously, such effects are not unique to condensate-forming proteins and have been documented for other protein classes as well (Gameiro et al, 2024; Weill et al, 2019).

These considerations are also relevant when comparing wild-type and mutant proteins. Since point mutations typically do not substantially alter surface properties or condensation behavior, they are generally expected to be influenced by tagging in a similar way as the wild-type POI. Truncation mutants often exhibit more pronounced changes in molecular and phase-separation properties, necessitating careful evaluation of both tag choice and position.

It is important to note that tagging can strongly influence the condensation of some POIs while leaving others virtually unaffected, even within the same condensate. For example, DDX3X and G3BP1 are both SG marker proteins, but in our experiments only DDX3X is sensitive to tagging, while G3BP1 is not. We observed a similar discrepancy between the PB marker proteins DDX6, which is minimally affected by tagging, and EDC3, which is strongly affected. Since concentrations of all molecules in a multi-component condensate are linked (Ruff et al, 2021), perturbing the concentration or partitioning of one component may affect the concentrations of all others, thus potentially changing the entire composition.

In general FP tags showed much stronger effects than peptide tags. EGFP and EYFP tend to promote condensate formation or may be even aggregation, likely in part caused by their dimerization propensity. Also, ligand-binding tags such as Halo or SNAP, particularly when labeled with their respective dyes, seem to induce a more aggregated appearance. In contrast, mCherry and mScarlet-

I strongly reduce condensation. Most FP tags from the green wavelength spectrum used in this study (EBFP2, ECFP, EGFP, mEGFP, EYFP, mYPet, mTurquoise2) are derived from the same original protein, avGFP from *Aequorea victoria*, and overall behave very similarly. In contrast, the FP tags from the red wavelength spectrum (mCherry, mScarlet-I, mRuby3, mApple, mKO) are derived from different original FPs (Appendix Figs. S1 and S2), potentially explaining their more individualistic characteristics.

Given the complex differences observed across various FP tags, it is not surprising that they do not correlate with any single easily measured biophysical property, such as size, isoelectric point or overall charge at a given pH (Table EV1). This suggests that multiple and/or more complex factors are involved. Our coarse-grained simulations support the idea that particularly important factors are the net charge and charge distribution on the protein surface for both the POI and the FP. Surface charge distribution has previously been described to influence condensation (Kim et al, 2024; Taneja and Holehouse, 2021), which is further supported by experiments demonstrating that charged peptide tags can drive and strongly influence condensation in cells and in vitro (Crabtree et al, 2023; Yeong et al, 2022). Our simulations suggest that electrostatic repulsion and attraction between the fusion proteins, and the energetic cost of bringing a charged protein into the dense phase, are important factors determining to what extent FPs modulate condensation. Since surface charge is to some extent pH-dependent, this may explain the strong pH sensitivity observed for certain FPs like mCherry. In this context, it is important to consider that at least in bacteria and yeast the intracellular pH typically becomes more acidic under stress conditions, dropping from ~7.4 to ~6.4 (Buchanan and Edelson, 1996; Martínez-Muñoz and Kane, 2008; Orij et al, 2009). Additionally, various condensates may establish complex gradients of pH and other ions (King et al, 2024; Posey et al, 2024) that may in turn both influence and be influenced by protein tagging. Consequently, the appearance of stress-induced condensate like PBs or SGs may be distorted if the POI is tagged with a FP that alters condensate structures in a pH-dependent manner.

## Recommendations for experimental design

The impact of protein tags on condensate appearance, as outlined in this study and described by others (Avecilla et al, 2025; Barkley et al, 2024; Dao et al, 2024; Fatti et al, 2025; Feric et al, 2016; Hernandez-Candia et al, 2024; Martin et al, 2021; Pandey et al, 2024; Uebel and Phillips, 2019; Zhou et al, 2025; Ziaunys et al, 2024), carries significant implications for experimental design and interpretation. While there is no one-fits-all solution, we try to give some overall recommendations.

For experiments in cells, we recommend comparing tagged constructs to immunostaining of the endogenous protein when

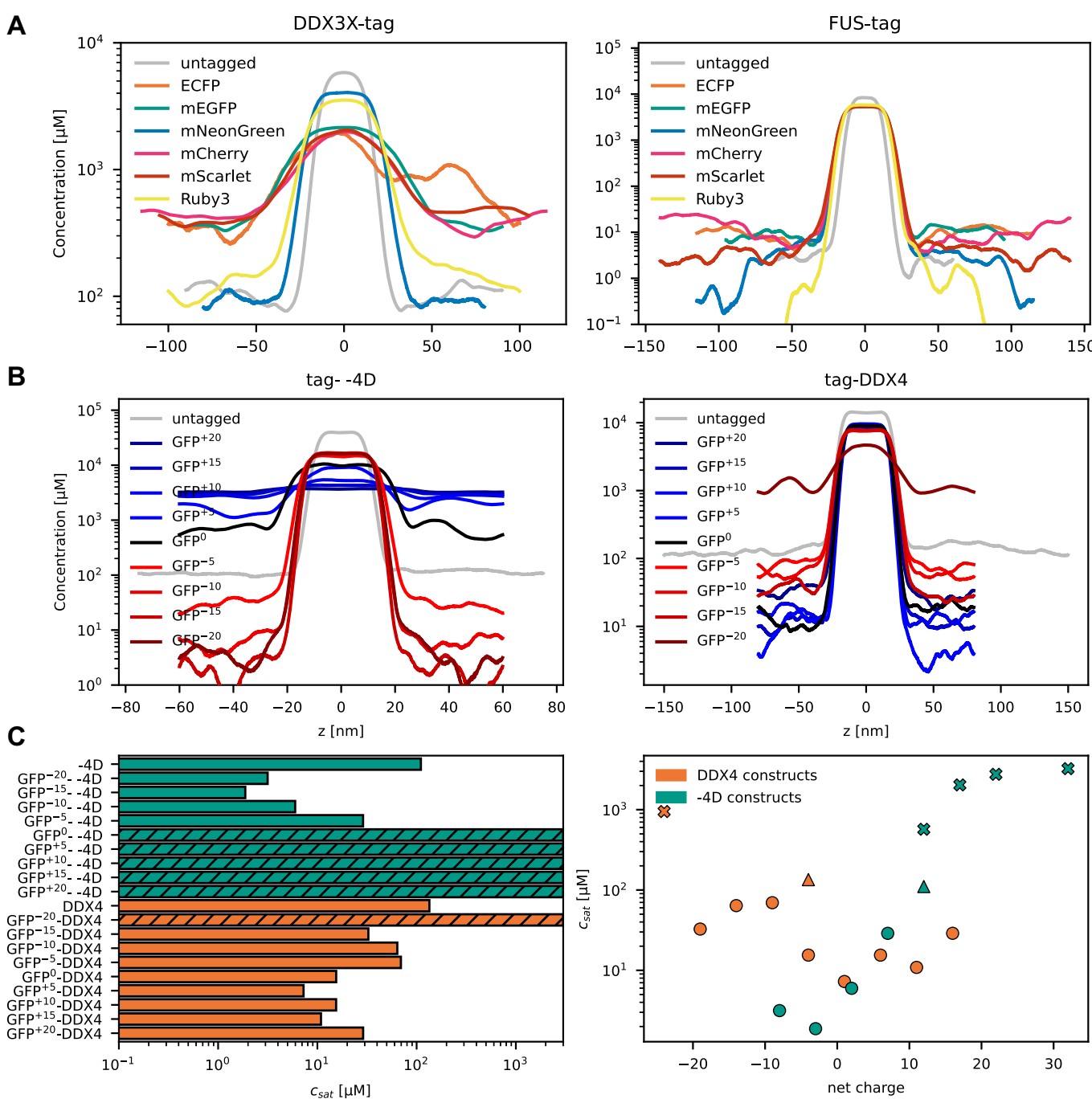

**Figure 6. Net charge of fluorescent protein tags influences condensation behavior in silico.**

Phase coexistence simulations were performed at 293 K with an ionic strength of 150 mM using CALVADOS 3. (**A**) Equilibrium density profiles of six FPs C-terminally fused with full-length DDX3X (left) and full-length FUS (right). (**B**) Equilibrium density profiles of the fully intrinsically disordered proteins -4D (left) and truncated DDX4 (right) N-terminally fused with GFP variants with varying net charge based with non-fluorescent GFP[S65G,Y66G] as GFP[0]. (**C**) Simulated saturation concentrations ($c_{sat}$, left) and correlation between $c_{sat}$ and net charge of for DDX4 (orange) or -4D (green) constructs with N-terminal tags. Hatched bars (left) and crosses (right) indicate systems that do not form stable condensates, and the values reported should be taken to mean that the $c_{sat}$ values are high but uncertain. In the right panel, circles represent tagged proteins, and the two triangles represent the untagged IDRs of DDX4 and the -4D variant of A1 LCD. Source data are available online for this figure.

POI-specific antibodies are available. If no POI-specific antibody is available, antibodies against other marker proteins can be used to determine changes in overall condensate appearance, or different tags can be compared. Testing the effects of tags on condensation in vitro before transitioning to cellular studies can also provide valuable insights, although additional cellular parameters, including the condensate environment, may modulate the response.

As is often the case in cellular work, overexpression of fusion proteins can artificially amplify or obscure condensation-related problems. Endogenous tagging or stable cell lines where POI levels can be titrated to match endogenous protein levels, can help to reduce variability and minimize artifacts. This approach aligns with recent studies which highlight that protein crowding and over-expression can mask or exaggerate the true phase-separation behavior of tagged proteins (Zhou et al, 2025).

For in vitro studies, we recommend avoiding FP tags as much as possible. Untagged recombinant proteins can be observed using brightfield microscopy, which poses diverse experimental challenges and limitations, or, more quantitatively, using quantitative phase microscopy (McCall et al, 2025). Therefore, we suggest spiking in substoichiometric amounts of the POI that has been chemically labeled at defined sites (e.g. the N-terminus, or cysteines) with bright and pH stable dyes such as ATTO. In our experience, these dyes cause minimal interference.

Overall, FP choice is typically not straightforward. In the end, many factors, including the type of application, technical limitations of instruments, phototoxicity, and compatibility with other markers will guide the best choice. Careful tag selection, thorough validation with appropriate controls, and systematic analysis are key to mitigate tag-related issues and gain accurate insights into condensate behavior.

While these challenges pose a concern for many experiments, they could also become an opportunity: switching FP tags, is an opportunity to intentionally modulate condensate material properties (Hernandez-Candia et al, 2024) and shift proteins to a more or less condensed, or even aggregated state, an approach that could offer valuable insights into the function of condensates.

# Methods

### Reagents and tools table

| Reagent/resource | Reference or source | Identifier or catalog number |
|---|---|---|
| **Experimental models** | | |
| **Cell lines** | | |
| HeLa K (HeLa Kyoto) | Kind gift from Stefanie Jonas lab | RRID:CVCL_1922 |
| HeLa FRT | Gromeier lab; Kaiser et al, 2008 | N/A |
| HeLa FRT DDX3X-EGFP | This study | N/A |
| HeLa FRT DDX3X-mEGFP | This study | N/A |
| HeLa FRT DDX3X-EYFP | This study | N/A |
| HeLa FRT DDX3X-mCherry | This study | N/A |

| Reagent/resource | Reference or source | Identifier or catalog number |
|---|---|---|
| HeLa FRT DDX3X-myc | This study | N/A |
| HeLa FRT DDX3X-V5 | This study | N/A |
| HeLa FRT EGFP-EDC3 | This study | N/A |
| HeLa FRT mEGFP-EDC3 | This study | N/A |
| HeLa FRT EYFP-EDC3 | This study | N/A |
| HeLa FRT mCherry-EDC3 | This study | N/A |
| **Bacterial strains** | | |
| One Shot™ TOP10 Chemically Competent E. coli | Thermo Fisher Scientific | Cat# C404003 |
| Lemo21(DE3) Competent E. coli | New England Biolabs | Cat# C2528H |
| **S. cerevisiae strains** | | |
| W303 PAB1-mScarlet::TRP1 (ade2-1 can1-100 GAL+ phi+ his3-11,15 ura3-1 leu2-3,112 trp1-1) | This study | N/A |
| W303 PAB1-mEGFP::KANMX (leu2-3,112 trp1-1 can1-100 ura3-1 ade2-1 his3-11,15 [phi + ]) | This study | N/A |
| W303 PAB1-mCherry::KANMX (leu2-3,112 trp1-1 can1-100 ura3-1 ade2-1 his3-11,15 [phi + ]) | This study | N/A |
| W303 PAB1-ECFP::KANMX (leu2-3,112 trp1-1 can1-100 ura3-1 ade2-1 his3-11,15 [phi + ]) | This study | N/A |
| W303 PAB1-YPet::KANMX (leu2-3,112 trp1-1 can1-100 ura3-1 ade2-1 his3-11,15 [phi + ]) | This study | N/A |
| W303 PAB1-mTurquoise::KANMX (leu2-3,112 trp1-1 can1-100 ura3-1 ade2-1 his3-11,15 [phi + ]) | This study | N/A |
| W303 PAB1-EYFP::KANMX (leu2-3,112 trp1-1 can1-100 ura3-1 ade2-1 his3-11,15 [phi + ]) | This study | N/A |
| BY PAB1-EGFP::HIS3 (leu2-3,112 trp1-1 can1-100 ura3-1 ade2-1 his3-11,15 [phi + ]) | This study | N/A |
| BY PAB1-mEGFP::KANMX (his3Δ1 leu2Δ0 lys2Δ0 ura3Δ0) | This study | N/A |
| BY PAB1-mCherry::KANMX (his3Δ1 leu2Δ0 lys2Δ0 ura3Δ0) | This study | N/A |
| BY PAB1-ECFP::KANMX (his3Δ1 leu2Δ0 lys2Δ0 ura3Δ0) | This study | N/A |
| BY PAB1-YPet::KANMX (his3Δ1 leu2Δ0 lys2Δ0 ura3Δ0) | This study | N/A |
| BY PAB1-mTurquoise::KANMX (his3Δ1 leu2Δ0 lys2Δ0 ura3Δ0) | This study | N/A |
| BY PAB1-EYFP::KANMX (his3Δ1 leu2Δ0 lys2Δ0 ura3Δ0) | This study | N/A |
| **Recombinant DNA** | | |
| pETMCN_His10-TS-3C_DDX3X-mCherry | This study | N/A |
| pETMCN_His10-TS-3C_DDX3X-mEGFP(A206K) | This study | N/A |

| Reagent/resource | Reference or source | Identifier or catalog number |
|---|---|---|
| pETMCN_His10-TS_3 C_DDX3X-EGFP | This study | N/A |
| pETMCN_His10-TS_3 C_DDX3X-EYFP | This study | N/A |
| pETMCN_His10-TS-3C_DDX3X-ECFP | This study | N/A |
| pETMCN_His10-MBP-3C-DDX3X | This study | N/A |
| pETMCN_His10-TS-3C-MPB-3C_CsdA-YPET | This study | N/A |
| pETMCN_His10-TS-3C-MPB-3C_CsdA-mNeonGreen | This study | N/A |
| pETMCN_His10-TS-3C-MPB-3C_CsdA-mKO2 | This study | N/A |
| pETMCN_His10-TS-3C-MPB-3C_CsdA-mApple | This study | N/A |
| pETMCN_His10-TS-3C-MPB-3C_CsdA-mTurquoise | This study | N/A |
| pETMCN_His10-MPB-3C_CsdA-mCherry | This study | N/A |
| pETMCN_His10-TS-3C-MPB-3C_CsdA-mScarlet-I | This study | N/A |
| pETMCN_His10-TS-3C-MPB-3C_CsdA-EBFP2 | This study | N/A |
| pETMCN_His10-TS-MPB-3C_CsdA-mEGFP(A206K) | This study | N/A |
| pETMCN_His10-TS-3C_CsdA_untagged | This study | N/A |
| pETMCN_His10-TS-3C_mCherry | This study | N/A |
| pETMCN_His10-TS-3C_mEGFP(A206K) | This study | N/A |
| pETMCN_His10-TS-3C_EGFP | This study | N/A |
| pETMCN_His10-TS-3C_EYFP | This study | N/A |
| pETMCN_His10-TS-3C_ECFP | This study | N/A |
| pETMCN_His10-MBP-3C_Pab1-EYFP | This study | N/A |
| pETMCN_His10-MBP-3C_Pab1-mEGFP(A206K) | This study | N/A |
| pETMCN_His10-MBP-3C_Pab1-mCherry | This study | N/A |
| pETMCN_His10-MBP-3C_Pab1-EGFP | This study | N/A |
| pETMCN_His10-MBP-3C_Pab1_untagged | This study | N/A |
| pETMCN_His10MBP_3C_FUS-EYFP | This study | N/A |
| pETMCN_His10MBP_3C_FUS-ECFP | This study | N/A |
| pETMCN_His10MBP_3C_FUS-mEGFP(A206K) | This study | N/A |
| pETMCN_His10MBP_3C_FUS-mCherry | This study | N/A |
| pETMCN_His10MBP_3C_FUS-EGFP | This study | N/A |

| Reagent/resource | Reference or source | Identifier or catalog number |
|---|---|---|
| pETMCN_His10-MBP-3C_FUS_untagged | This study | N/A |
| pETMCN_His6-TS-3C_Nab2-ymEGFP | This study | N/A |
| pcDNA5_FRTTO_DDX3X-SNAP | This study | N/A |
| pcDNA5_FRTTO_DDX3X-YPet | This study | N/A |
| pcDNA5_FRTTO_DDX3X-V5 | This study | N/A |
| pcDNA5_FRTTO_DDX3X-mNeonGreen | This study | N/A |
| pcDNA5_FRTTO_DDX3X-ECFP | This study | N/A |
| pcDNA5_FRTTO_DDX3X-EYFP | This study | N/A |
| pcDNA5_FRTTO_DDX3X-Halo | This study | N/A |
| pcDNA5_FRTTO_DDX3X-mScarlet-I | This study | N/A |
| pcDNA5_FRTTO_DDX3X-FLAG | This study | N/A |
| pcDNA5_FRTTO_DDX3X-HA | This study | N/A |
| pcDNA5_FRTTO_DDX3X-ALFA | This study | N/A |
| pcDNA5_FRTTO_DDX3X-EGFP | This study | N/A |
| pcDNA5_FRTTO_DDX3X-mEGFP(A206K) | This study | N/A |
| pcDNA5_FRTTO_DDX3X-myc | This study | N/A |
| pcDNA5_FRTTO_DDX3X-mCherry | This study | N/A |
| pcDNA5_FRTTO_mCherry-G3BP1 | This study | N/A |
| pcDNA5_FRTTO_DDX6-mCherry | This study | N/A |
| pcDNA5_FRTTO_DDX6-mEGFP | This study | N/A |
| pcDNA5_FRTTO_DDX6-V5 | This study | N/A |
| pcDNA5_FRTTO_mEGFP(A206K)-G3BP1 | This study | N/A |
| pcDNA5_FRTTO_EYFP-G3BP1 | This study | N/A |
| pcDNA5_FRTTO_V5-G3BP1 | This study | N/A |
| pcDNA5_FRTTO_myc-G3BP1 | This study | N/A |
| pcDNA5_FRTTO_EGFP-G3BP1 | This study | N/A |
| pcDNA5_FRTTO_V5-EDC3 | This study | N/A |
| pcDNA5_FRTTO_EGFP-EDC3 | This study | N/A |
| pcDNA5_FRTTO_mEGFP(A206K)-EDC3 | This study | N/A |
| pcDNA5_FRTTO_mCherry-EDC3 | This study | N/A |
| pcDNA5_FRTTO_EYFP-EDC3 | This study | N/A |
| pcDNA5_FRTTO_myc-EDC3 | This study | N/A |
| pOG44 | Thermo Fisher Scientific | Cat# V600520 |
| pcDNA5_FRTTO_SRRM2-EYFP | This study | N/A |
| pcDNA5_FRTTO_SRRM2-mCherry | This study | N/A |
| pcDNA5_FRTTO_SRRM2-mEGFP(A206K) | This study | N/A |
| pcDNA5_FRTTO_SRRM2-EGFP | This study | N/A |

| Reagent/resource | Reference or source | Identifier or catalog number |
|---|---|---|
| **Antibodies** | | |
| Anti-β-actin, C4 (mouse monoclonal) | Sigma | Cat# MAB1501; RRID:AB_2223041 |
| Anti-myc (mouse monoclonal) | Genetex | Cat# GTX29106; RRID:AB_369669 |
| Anti-V5 (mouse monoclonal) | Genetex | Cat# GTX635260; RRID: AB_2888514 |
| Anti-HA (rabbit polyclonal) | Abcam | Cat# ab20084; RRID:AB_445319 |
| Anti-FLAG (mouse monoclonal) | Cell Signaling | Cat# 8146; RRID:AB_10950495 |
| Atto488 FluoTag-X2 anti-ALFA (1G5) | Nanotag Biotech | Cat# N1502-At488-S; RRID: AB_3075982 |
| Anti-DDX3X (mouse monoclonal) | Abcam | Cat# ab196032; RRID: AB_2910197 |
| anti-DDX6 (rabbit polyclonal) | Abcam | Cat# ab70455; RRID: AB_1209637 |
| Anti-G3BP1 (rabbit polyclonal) | Genetex | Cat# GTX112191; RRID: AB_11164802 |
| Anti-EDC3 (mouse monoclonal) | Santa Cruz | Cat# sc-365024; RRID: AB_10841422 |
| Anti-DCP1A (rabbit, monoclonal) | Abcam | Cat# ab183709, RRID: AB_3331658 |
| Anti-SRRM2 ("SC35", mouse monoclonal) | Genetex | Cat# GTX11826; RRID:AB_372954 |
| Anti-Halo (rabbit, polyclonal) | Promega | Cat# G9281, RRID:AB_713650 |
| Anti-SNAP (rabbit, polyclonal) | New England Biolabs | Cat# P9310S, RRID:AB_10631145 |
| Anti-SON (rabbit, polyclonal) | Genetex | Cat# GTX129778, RRID:AB_2886086 |
| Alexa Fluor 488 goat anti-rabbit IgG (H + L) | Thermo Fisher Scientific | Cat# A-11008; RRID:AB_143165 |
| Alexa Fluor 568 goat anti-rabbit IgG (H + L) | Thermo Fisher Scientific | Cat# A-11011; RRID:AB_143157 |
| Alexa Fluor 647 goat anti-rabbit IgG (H + L) | Thermo Fisher Scientific | Cat# A-21244; RRID:AB_2535812 |
| Alexa Fluor 488 goat anti-mouse IgG (H + L) | Thermo Fisher Scientific | Cat# A-11001; RRID:AB_2534069 |
| **Chemicals, enzymes and other reagents** | | |
| FuGeneHD | Promega | Cat# E2311 |
| SNAP-Surface 549 | NEB | Cat# S9112S |
| Halo TMR ligand | Promega | Cat# G8251 |
| Vectashield | Vector Laboratories | Cat# H-1200-10 |
| ATTO488 NHS-ester dyes | ATTO-TEC GmbH | Cat# AD 488-31 |
| PMSF Protease inhibitors | Carl Roth | Cat# 6367.3xxx |
| DNaseI | Roche | Cat# 10104159001 |

| Reagent/resource | Reference or source | Identifier or catalog number |
|---|---|---|
| RNaseA | Macherey-Nagel | Cat# 74050 |
| AcuteBand Pre-Stained Protein Ladder | Lubio | Cat# LU5001-2500 |
| Precission (3C) protease | GeneScript | Cat# Z02799 |
| Poly(A) | Sigma | Cat# P9403-25 mg |
| Poly(U) | Sigma | Cat# P9528-25 mg |
| Strep Tactin conjugated to HRP | iba | Cat# 2-1502-001 |
| **Software** | | |
| OMERO | Allan et al, 2012 | https://www.openmicroscopy.org/ |
| Cell Profiler | Stirling et al, 2021 | https://cellprofiler.org/ |
| Fiji | Schindelin et al, 2012 | https://fiji.sc/; RRID: SCR_002285 |
| Fiji adaptiveThr function | ImageJ plugin by Quingzong Tseng | https://sites.google.com/site/qingzongtseng/imagejplugins |
| EasyFRAP | Koulouras et al, 2018 | https://easyfrap.vmnet.upatras.gr/ |
| R Studio | Posit team | https://posit.co/download/rstudio-desktop/ |
| Adobe Photoshop | Adobe | https://www.adobe.com/ |
| Adobe Illustrator | Adobe | https://www.adobe.com/ |
| AlphaFold v2.3 | Jumper et al, 2021 | |
| Prism 8 | GraphPad Software | https://graphpad.com/features; RRID: SCR_002798 |
| FPbase | Lambert (2019) | https://www.fpbase.org/ |
| Sedfit | Schuck (2000 | |
| StackReg plugin | Thevenaz et al, 1998 | https://bigwww.epfl.ch/thevenaz/stackreg/ |
| CALVADOS3 | Cao et al, 2024 | |
| Fuzdrop | Hatos et al, 2022 | |
| **Other** | | |
| 384-well plate for condensation assays: black, optically clear flat-bottom, ultra-low-attachment-coated PhenoPlates | Revvity | Cat# 6057800 |
| 384 well plate for yeast imaging | Brooks | Cat# MGB101-1-2-LG-L |

## Human cell culture and cell lines

Human cell lines used in this study are listed in the Reagents and Tools table. All cell lines were cultured at 37 °C and 5% $CO_2$ in

DMEM supplemented with 10% fetal calf serum and 0.1 mg/mL penicillin/streptomycin. Tetracycline inducible cell lines were generated based on the FlpIn system (Invitrogen) allowing stable integration of pcDNA5-based constructs. Parental Hela cells containing an FRT site (Kaiser et al, 2008), were used to obtain monoclonal cell lines after selection with 0.3 mg/mL hygromycin B. All cell lines were tested negative for mycoplasma using PCR-based testing. Construct expression was induced with doxycycline.

## Yeast cell construction and culturing

*Saccharomyces cerevisiae* strains used in this study are listed in the Reagents and Tools table. Strains were constructed by transformation of a PCR amplification product with homology to the target site as described previously (Longtine et al, 1998). Cells were cultured in synthetic complete medium containing 2% D-glucose at 30 °C.

## Bacteria strains and culturing

*Escherichia coli* were cultured at 37 °C in lysogeny broth (LB) medium with agitation at 150–200 rpm or on agar plates. Bacterial cultures carrying plasmids were supplemented with the corresponding antibiotic selection in the following concentrations; 50 µg/mL kanamycin, 100 µg/mL ampicillin or 30 µg/mL chloramphenicol. Cloning and plasmid amplification were performed in *E. coli* Top10 and protein production carried out in *E. coli* Lemo21(DE3). Plasmids used in this study are listed in the Reagents and Tools table.

## Plasmid construction

All proteins for expression in human cells were cloned into the pcDNA5 vector, with a GSGGSGG linker between the POI and the FP or peptide tag. Constructs for recombinant expression in *E. coli* and purification were cloned into pETMCN vectors, with the same GSGGSGG linker between POI and FP.

## Transient transfection, drug treatment and fixation of human cells

HeLa K cells grown on coverslips were transfected with plasmid DNA (listed in the Reagents and Tools table) using FuGeneHD transfection reagent (Promega, Cat# E2311). 24 h after transfection cells were fixed with 3% paraformaldehyde in PB for 15 min at room temperature, where indicated cells were treated with 500 µM sodium arsenite for 30 min prior to fixation to induce SG or PB formation. To label SNAP or Halo tagged proteins in cells, cells were incubated with either 2 µM SNAP-Surface 549 dye (NEB, Cat# S9112S) for 30 min before fixation or 200 nM TMR ligand (Promega, Cat# G8251) for 15 min and washed three times with PB before fixation.

## Immunostaining, microscopy of fixed human cells, live cell imaging

After fixation with paraformaldehyde, cells were permeabilized with 0.1% Triton X-100, 0.02% SDS in PBS for 7 min. Alternatively cells were fixed with −20 °C Methanol for 6 min at −20 °C.

Epitopes were then blocked with 2% BSA in PBS (BSA/PB) for 30 min. To stain the protein of interest, cells were next incubated with primary antibody (listed in the Reagents and Tools table) in BSA/PBS for 1 h. After three washes with PBS, cells were incubated with the respective secondary antibody diluted in BSA/PBS for 30 min. Following three washes with PBS, coverslips were mounted onto glass slides with Vectashield for confocal microscopy. Confocal images were acquired either using a Zeiss LSM700 Upright microscope with a 63×1.4 NA oil DIC Plan Apochromat objective or a Nikon Ti2 CrEST V3 spinning disk microscope with a 100 × 1.45 NA CFI Plan Apochromat Lambda D oil objective. For live cell imaging, cells were seeded into µ-slide 8-well ibidi chambers. Images were acquired with a Nikon Ti2 CrEST V3 spinning disk microscope using a 100 × 1.45 NA CFI Plan Apochromat Lambda D oil objective in a 5% $CO_2$, 37 °C chamber. In all panels, representative maximum Z-projections are shown.

## Immunoblot analysis

Cells were directly harvested with SDS sample buffer, boiled at 95 °C for 5 min. Samples were separated on SDS-PAGE gels and proteins were transferred onto an activated PVDF membrane by wet blotting. The membrane was blocked with 4% milk in PBST for 30 min, before it was incubated with primary antibody diluted in 4% milk in PBS overnight at 4 °C. After three washes for 5 min with PBST, the membrane was incubated in 4% milk containing the respective secondary antibody fused with HRP. Again, the membrane was washed three times and developed with ECL. The signal was detected using a Fusion (Vilber) imaging system.

## Protein expression and purification

Recombinant proteins were expressed in *E. coli Lemo21(DE3)* by transforming chemically competent cells with the appropriate pETMCN-based expression plasmids under the respective antibiotic selection. Pre-cultures were grown overnight at 37 °C in LB containing antibiotics and 1% D-glucose. Expression cultures were inoculated to yield an OD600 of 0.05 in Terrific Broth (TB) medium containing antibiotics and 1% D-glucose, grown at 37 °C to an $OD_{600}$ of 0.6–0.8 and induced with a final concentration of 200 mM IPTG. Cells were grown overnight at 18 °C and harvested by centrifugation (5000×g, 15 min, 4 °C). Cell lysis was carried out in 30 mL lysis buffer (25 mM Tris-HCl pH 7.5, 1 M NaCl, 2 mM $MgCl_2$, 10% glycerol (w/v), protease inhibitors, DNase (Roche, Cat# 10104159001), and RNase (Macherey-Nagel, Cat# 74050) per 1 L of expression culture. Cells were lysed by pressure homogenisation through an EmulsiFlex (Avestin) and/or sonication. Even if already lysed by Emulsiflex, sonication treatment can be critical to help with the removal of residual RNA after RNAse treatment (e.g. for FUS). Lysates were cleared by centrifugation (20,000×g, 30 min, 4 °C) and filtration through a 0.45 µm cut-off filter. His-tagged proteins were purified by immobilized metal affinity chromatography (IMAC) using self-made $Ni^{2+}$ Sepharose columns, followed by 3 C protease cleavage with simultaneous overnight dialysis. His10-TwinStrep-MBP tags and uncleaved proteins were removed by reverse IMAC. Proteins were further purified by size exclusion chromatography (SEC) using a 16/600 HiLoad Superdex 200 pg column (Cytiva) on an ÄKTA pure (GE Life Sciences) into the final storage buffer. IMAC, reverse IMAC and gel filtration fractions, as

well as final protein pools were analyzed by SDS-PAGE and Coomassie staining, for size reference AcuteBand Pre-Stained Protein Ladder (Lubio, Cat# LU5001-2500) was loaded. Clean SEC fractions were pooled and concentrated using Amicon Ultra-4 centrifugal filters (Merck Millipore). Concentrated protein aliquots were snap-frozen in liquid nitrogen and stored at −80 °C. This general protein expression and purification protocol was applied to all DDX3X, CsdA, Pab1 and Nab2 constructs as well as the sole fluorescent protein constructs with small adaptations, i.e. by adjusting the buffer pH according to pI of the protein. In case of the FUS constructs, the 10xHis-MBP tag was not cleaved off, and subsequently reverse IMAC was not performed. Lysis methods, purification steps and storage buffer information for each individual recombinant protein are listed in Table EV4.

## Chemical labeling of proteins

Untagged recombinant proteins were chemically labeled with ATTO488 NHS-ester dyes (ATTO-TEC GmbH, Cat# AD 488-31). To specifically label the primary amine of the N-terminus and not the lysines, labeling reactions were performed at pH 6.5 or lower, except for FUS, where labeling was performed at pH 7.5. If the protein storage buffer contained primary amines, buffer exchange into phosphate buffer was performed using Zeba Spin Desalting Columns, 7 K MWCO (Thermo Fisher Scientific) prior to reaction assembly. Protein and ATTO488 NHS-ester dye were mixed in a molar ratio of 1:4, followed by an incubation of 1 h at room temperature, protected from light. Using Zeba Spin Desalting Columns free dye was removed and labeled proteins were transferred back to the original protein storage buffer. Finally, labeled proteins were concentrated using Amicon Ultra-4 centrifugal filters (Merck Millipore) to 100–200 µM. Protein concentration and average degree of labeling was determined for each protein. Concentrated protein was aliquoted, snap-frozen in liquid nitrogen and stored at −80 °C. For microscopy-based assays labeled proteins were spiked-in, in a 1:100 molar ratio to the unlabeled protein, if not stated differently.

## In vitro condensation assay

Condensation assays were carried out in 384-well, black, optically clear flat-bottom, ultra-low-attachment-coated PhenoPlates (Revvity, Cat# 6057800), following the procedure previously described by Hondele and colleagues (Hondele et al, 2019). In short, protein stocks were diluted using protein storage buffers to the tenfold of the final assay concentration. For visualization of the untagged protein variants, 1% of ATTO488 labeled protein was spiked-in the tenfold concentrated protein stock. The tenfold protein solution was transferred to the corner of a 384-well, where it was mixed in a ratio of 1:10 with corresponding assay master mixes, yielding final assay conditions as listed for the individual proteins in Table EV5. Poly(U) (Cat# P9528-25 mg) and Poly(A) (Cat# P9403-25 mg) RNA were obtained from Sigma. After reaction assembly plate was centrifuged (10×g, 30 s, 25 °C). The plate was incubated on the stage of the temperature-controlled microscope (25 °C: DDX3X, FUS, CsdA and Nab2 or 30 °C: Pab1). Reaction assembly and imaging were conducted at coordinated time intervals, with a constant total incubation time maintained across all tested fluorescent protein constructs to allow for direct comparison. Images were acquired on a temperature-controlled, inverted Nikon Ti2 wide field microscope equipped with a ×40 Plan Apo Lambda air objective NA 0.95, a Lumencor SPECTRA light source and a Hamamatsu ORCA-Fusion cMOS camera. All conditions were imaged using Differential Interference Contrast (DIC) microscopy and the appropriate fluorescence channels for each respective FP. Four images were taken in each of the three independent replicates. The microscope was controlled with the Nikon NIS Elements software and automated using the integrated JOBS system. Light intensity and exposure time were adjusted to the different fluorescent protein tags and maintained consistent within one experiment, and might differ between replicates. Images were processed by adjusting brightness and contrast using the open microscopy environment (OMERO). For each panel, representative images are shown.

## Fluorescence recovery after photobleaching

Fluorescence recovery after photobleaching (FRAP) assays of in vitro condensates of DDX3X constructs (tagged or untagged + 1% ATTO488-DDX3X spike-in) were set up in 384-well, black, optically clear flat-bottom, ultra-low-attachment-coated Pheno-Plates. Protein stock and condensation triggering master mix solutions were prepared identical to the in vitro condensation assay described above. Final assay conditions: 25 mM sodium phosphate buffer at pH 6.2, 200 mM NaCl, 2 mM MgCl₂, 0.5 mg/mL BSA, 0.05 mg/mL poly(U), 0.5 mM ATP, 0.5 mM DTT. The concentrated protein solutions were transferred to the corner of a 384-well, where it was mixed in a ratio of 1:10 with the corresponding assay master mix. Plate was centrifuged (10×g, 60 s, 25 °C), allowing reconstituted droplets to settle, followed by a total incubation of 1 h at room temperature prior to FRAP acquisition. Photobleaching and image acquisition were performed on an inverted Olympus SpinD (CSU-W1) spinning disc microscope equipped with a RAPP UGA-42 Firefly 3 L Photomanipulation/FRAP module, a 60x UPL APO oil objective NA 1.5 and a Hamamatsu ORCA-Fusion sCMOS camera. A circular region of interests (ROI) was bleached using a DL-473/100 laser. FRAP ROIs and laser intensity were controlled by SysCon2.0 software. The general FRAP experiment scheme included the acquisition of three pre-bleach frames, prior photobleaching, followed by 180 post-bleach frames in a 1 s time interval. Standard deviation was assessed by comparing individual droplets within and across replicates. Images of DDX3X-mCherry were acquired using a 561 nm diode laser in combination with a 580–653 nm single bandpass filter. Images of DDX3X-mEGFP, DDX3X-EGFP, untagged DDX3X + 1% ATTO488-DDX3X spike-in and DDX3X-EYFP were acquired using a 488 nm diode laser in combination with a 500-550 nm single bandpass filter. Microscope and image acquisition were controlled with the Olympus cellSens software.

## Analytical ultracentrifugation

Sedimentation velocity experiments were performed in 25 mM phosphate buffer pH 6.2 or 7.4, 150 mM NaCl, 1% glycerol, 1 mM MgCl₂ and 0.5 mM β-mercaptoethanol, on 410 µL samples in double sector charcoal-epon centerpieces at 42,000 rpm and 20 °C using a Beckman XL-I analytical ultracentrifuge with the Beckman An-50 Ti rotor. Sedimentation was monitored during an overnight

run using the interference detector with 900 scans and a nominal interscan delay of 1 min. The buffer densities (pH 6.2: 1.0105 g/ml, pH 7.4: 1.0098 g/ml) and viscosities (pH 6.2: 1.0214 centipoise, pH 7.4: 1.0443 centipoise) were measured at 20 °C using an Anton Paar DMA 4500 M density meter and AMVn viscometer.

## Microscopy of glycerol stressed yeast cells

Cells were inoculated from saturated overnight cultures into fresh synthetic media with the respective amino acid selection to $OD_{600}$ 0.05-0.1, and grown at 30 °C to $OD_{600}$ 0.6–0.8. Cells were then transferred to a 384-well plate (Brooks, Matrical MGB101-1-2-LG-L) treated with Concanavalin A (stock solution of 1 mg/ml, air-dried), and spun down (100×$g$, 1 min, 20 °C). For acute glycerol stress, Pab1-tagged strains were washed three times in synthetic medium without glucose supplemented with final 3% glycerol directly in the imaging well, resuspended in the same medium and cultured for 45 min prior to imaging. All imaging experiments were performed at 30 °C. Microscopy was performed using an inverted wide field microscope (Nikon Ti2) equipped with a Lumencor SPECTRA light source and a Hamamatsu ORCA-Fusion cMOS camera using a 60x Plan-Apo objective NA 1.4 and NIS Elements software.

## Simulations

We generated fusion protein sequences by attaching tags to either the N-terminus or C-terminus of a POI. In this study, we investigated four POI: two full-length proteins containing folded domains, DDX3X (FL_DDX3X) and FUS (FL_FUS), and two intrinsically disordered proteins without folded domains, -4D (Bremer et al, 2022) and the isolated IDR of DDX4 (Brady et al, 2017). For FL_DDX3X and FL_FUS, we used AlphaFold to predict the structures of the fusion proteins. These predictions were then aligned with the published structures of the folded domains of DDX3X and FUS (catalytic core of DDX3X (PDB CODE: 5E7I), FUS-ZnF (PDB CODE: 6G99), FUS-RRM domain (PDB CODE: 2LA6)) to generate superimposed structures. The superimposed folded domains were then combined again with the IDRs from the AlphaFold predictions. Following relaxation of these structures, we used them as input for subsequent coarse-grained (CG) simulations. We used the following tags for simulations: ECFP, mEGFP, mNeonGreen, mCherry, mScarlet, Ruby3 as well as GFP variants with different charges (Taneja and Holehouse, 2021), $GFP^{+20}$, $GFP^{+15}$, $GFP^{+10}$, $GFP^{+5}$, $GFP^{-5}$, $GFP^{-10}$, $GFP^{-15}$, and $GFP^{-20}$ based on non-fluorescent $GFP^{S65G,Y66G}$ as $GFP^0$.

We applied the previously developed coarse-grained CALVA-DOS 3 model (Cao et al, 2024) to perform direct-coexistence simulations in a cuboidal box. Each residue was represented as a single bead. $C_\alpha$ positions were used for intrinsically disordered regions (IDRs) while center-of-mass position of residues were used for folded domains. The potentials of the model consist of a harmonic potential for chain connectivity, Ashbaugh-Hatch potential (Ashbaugh and Hatch, 2008) and Debye-Hückel potential (Dignon et al, 2018) for non-bonded interactions and another harmonic potential for restraining non-bonded residue pairs within the same folded domain. We simulated 100 chains for each fusion protein. To avoid steric clashes of densely packed input structures, we selected the most compact conformation sampled by single-chain simulations with CALVADOS 3 as the initial

conformation for each multi-domain protein (MDP) chain. Before production simulation, we performed equilibrium runs where an external force was used to push chains towards the box center so that a condensate could be formed. We then started production runs, saved frames every 0.125 ns and discarded the first 150 ns before analysis. The slab in every frame was centered in the box and the equilibrium density profile $\rho(z)$ was calculated by taking the averaged densities over the trajectories as previously described (Tesei and Lindorff-Larsen, 2022). Errors of the saturation concentrations reported in Table EV3 correspond to standard errors estimated using the "BLOCKING" method, in which trajectories were divided into optimally sized blocks to assess uncertainty (https://github.com/FrPsc/BLOCKING).

### Quantification and statistical analysis
All software and algorithms used in this study are listed in the Reagents and Tools table.

## Quantification of foci in human cells

Maximum Z-projections were performed using Fiji software (version 2.14.0/1.54 f). Cell profiler 2 (v 4.2.6) was employed to identify cells/nuclei and to segment DDX3X, EDC3, G3BP1, DCP1 and SRRM2 foci. After feature extraction, plots were generated using R Studio.

## Quantification of in vitro condensation assays

Cell Profiler 2 (v 4.2.6) was used for droplet segmentation and feature extraction. Mean droplet number per 0.1 mm², mean droplet area in μm², mean partition coefficient (PC) and mean image intensity standard deviation were quantified and plotted using R Studio. Please note that the partition coefficient (PC) was calculated based on widefield microscopy data, which may not provide optimal accuracy; confocal imaging would be more ideal for this purpose. Impact of fluorescent tagging with mEGFP and mCherry compared to untagged DDX3X, FUS and CsdA was assessed by calculating and plotting the mean log2 fold change (FP/untagged) of droplet number, droplet area, PC and image intensity standard deviation using R.

## Quantification of analytical ultra-centrifugation

The sedimentation velocity data were fitted to a diffusion-deconvoluted sedimentation coefficient distribution, c(s), using the software Sedfit 100 scans equally spaced in time and spanning the time period when the radial concentration profile was significantly changing were selected for analysis. A sedimentation coefficient range of 1–15 s was used with a resolution of 100 points. A single value of the frictional ratio ($f/f_0$) for all sedimenting species, radially invariant and time-invariant noise, baseline, meniscus and bottom position were all fitted.

## Quantification of Fluorescence recovery after photobleaching

For analyzing the FRAP data, individual droplets were stabilized using the StackReg plugin in Fiji software (version 2.14.0/1.54 f) for image alignment. Next, images were analyzed using the easyFRAP software to quantify fluorescence recovery dynamics.

## Quantification of foci in yeast cells

Images of yeast cells were processed using Fiji software (version 2.14.0/ 1.54 f) and plots were generated in R Studio. Granule quantification was performed using the Fiji adaptiveThr function (ImageJ plugin by Quingzong Tseng) with a weighted mean and strict thresholds to remove background and to isolate individual fluorescent foci.

## Statistical analysis

Summary statistics including calculation of mean, median, N, standard deviation, standard error of the mean as well as the Pearson correlation coefficient was performed using R. *T* tests were also performed in R as two-sided *t* tests. Sample size is indicated in the figure legend, where applicable.

# Data availability

Imaging data: BioImage Archive: S-BIAD2192.

The source data of this paper are collected in the following database record: biostudies:S-SCDT-10_1038-S44319-025-00626-y.

# Peer review information

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

## Acknowledgements

We would like to thank Anthony A. Hyman, Markus Affolter and members of the Hondele lab, particularly Giulia Basile, Masroor Kahloon and Arturo Sanchez for helpful discussions and critical reading of the manuscript. We thank Philippe van der Stappen for assistance with FRAP experiments. We thank Sebastian Abegg for assistance with protein purification. We thank the imaging facility (IMCF) and the biophysics facility of the Biozentrum for their support on this project. MG and DO were funded by "Fellowships for Excellence" from the International PhD Program in Molecular Life Sciences of the Biozentrum, University of Basel. MH is funded by the Swiss National Science Foundation (PCEFP3_187052) and the European Research Council (ERC-ST2020 950262). KL-L and FC are supported by the PRISM (Protein Interactions and Stability in Medicine and Genomics) centre funded by the Novo Nordisk Foundation (NNF18OC0033950, to KL-L), and the China Scholarship Council (CSC 202206340019, to FC).

## Author contributions

**Kerstin Dörner**: Conceptualization; Data curation; Formal analysis; Validation; Investigation; Visualization; Methodology; Writing—original draft; Project administration; Writing—review and editing. **Michelle Jennifer Gut**: Conceptualization; Data curation; Formal analysis; Supervision; Validation; Investigation; Visualization; Methodology; Writing—original draft; Writing—review and editing. **Daan Overwijn**: Conceptualization; Data curation; Formal analysis; Validation; Investigation; Visualization; Methodology; Writing—review and editing. **Fan Cao**: Data curation; Software; Formal analysis; Investigation; Visualization; Methodology; Writing—original draft; Writing—review and editing. **Matej Siketanc**: Data curation; Formal analysis; Validation; Investigation; Visualization; Methodology; Writing—review and editing. **Stephanie Heinrich**: Data curation; Formal analysis; Validation; Investigation; Visualization; Methodology; Writing—review and editing. **Nicole Beuret**: Validation; Investigation; Writing—review and editing. **Justin Meyer**: Validation; Investigation; Visualization. **Timothy Sharpe**: Investigation; Methodology; Writing—review and editing. **Kresten Lindorff-Larsen**: Conceptualization; Resources; Data curation; Software; Formal analysis; Supervision; Funding acquisition; Validation; Investigation; Visualization; Methodology; Writing—original draft; Project administration; Writing—review and editing. **Maria Hondele**: Conceptualization; Resources; Data curation; Software; Formal analysis; Supervision; Funding acquisition; Validation; Investigation; Visualization; Methodology; Writing—original draft; Project administration; Writing—review and editing.

Source data underlying figure panels in this paper may have individual authorship assigned. Where available, figure panel/source data authorship is listed in the following database record: biostudies:S-SCDT-10_1038-S44319-025-00626-y.

## Disclosure and competing interests statement

KL-L holds stock options in and is a consultant for Peptone Ltd. The remaining authors declare no competing interests.

# Expanded View Figures

Figure EV1. DDX3X tag effects are not correlated with expression level and evident in live cells.

(A, B) HeLa K cells were transiently transfected with DDX3X plasmids for 24 h, and stressed with 500 µM sodium arsenite for 30 min before fixation. Quantification of DDX3X foci as displayed in Fig. 1A: number (D) and area (E) relative to DDX3X-tag expression level. Pearson correlation coefficient (r) is indicated. $N = 3$, $n \geq 135$ cells. (C, D) Stable inducible HeLa cell lines were induced with doxycycline for 24 h to express the respective DDX3X-tag constructs. Cells were stressed with 500 µM sodium arsenite for 30 min and imaged live. Scale bar: 20 µm. Quantification of the number of DDX3X foci and their area (D). Box plots show the median (center line), 25th–75th percentiles (bounds of box), and whiskers=1.5x IQR. $N = 3$, $n \geq 110$ cells. (E, F) Stable inducible HeLa cell lines were induced with doxycycline for 24 h to express the respective DDX3X-tag constructs. Cells were stressed with 500 µM sodium arsenite for 30 min. Cells were also stained for DDX3X with immunostaining. Scale bar: 20 µm. Quantification of the number of FP-DDX3X and DDX3X foci detected by the DDX3X antibody (F). Box plots show the median (center line), 25th–75th percentiles (bounds of box), and whiskers=1.5x IQR. $N = 3$. $n \geq 40$ cells.

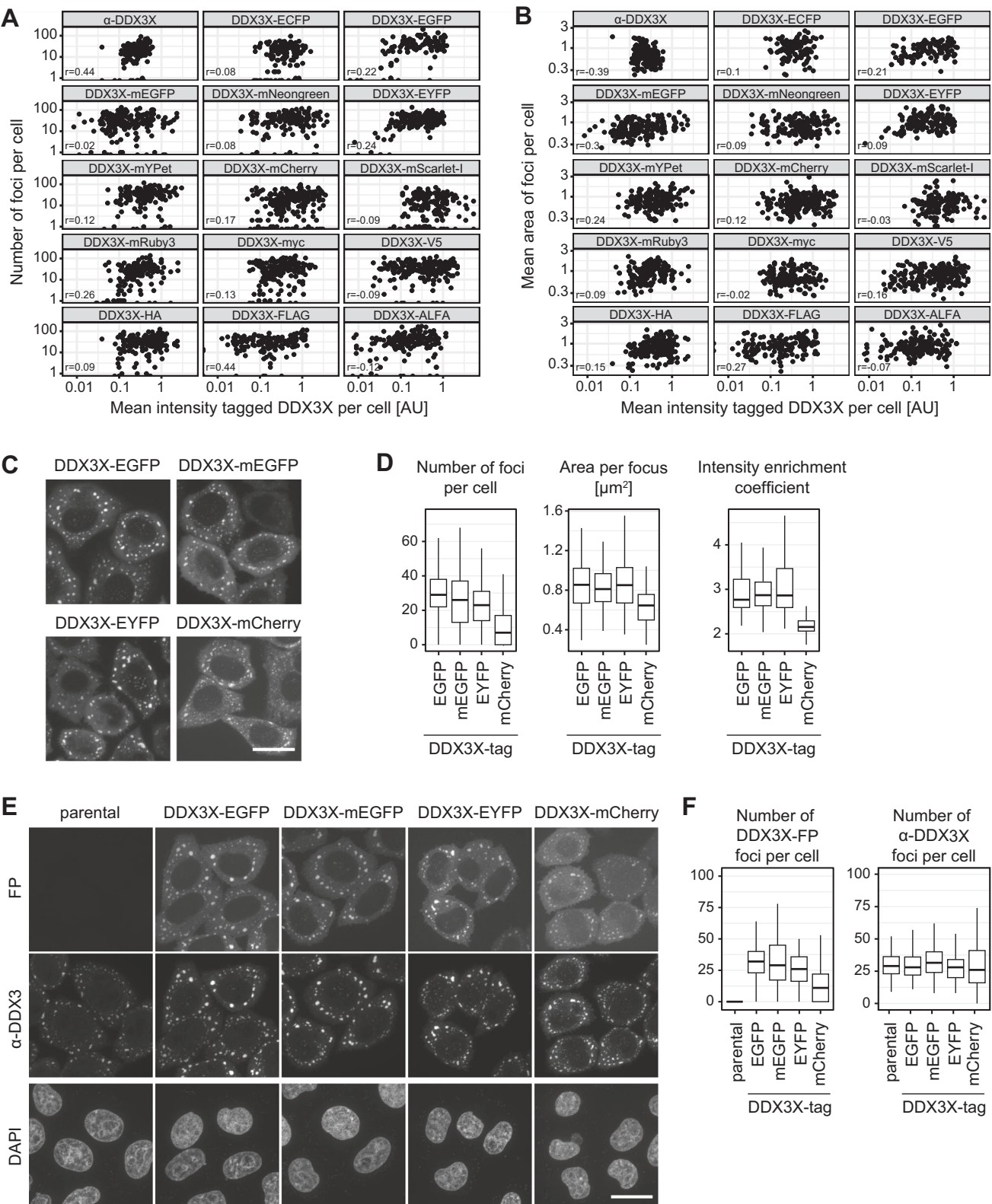

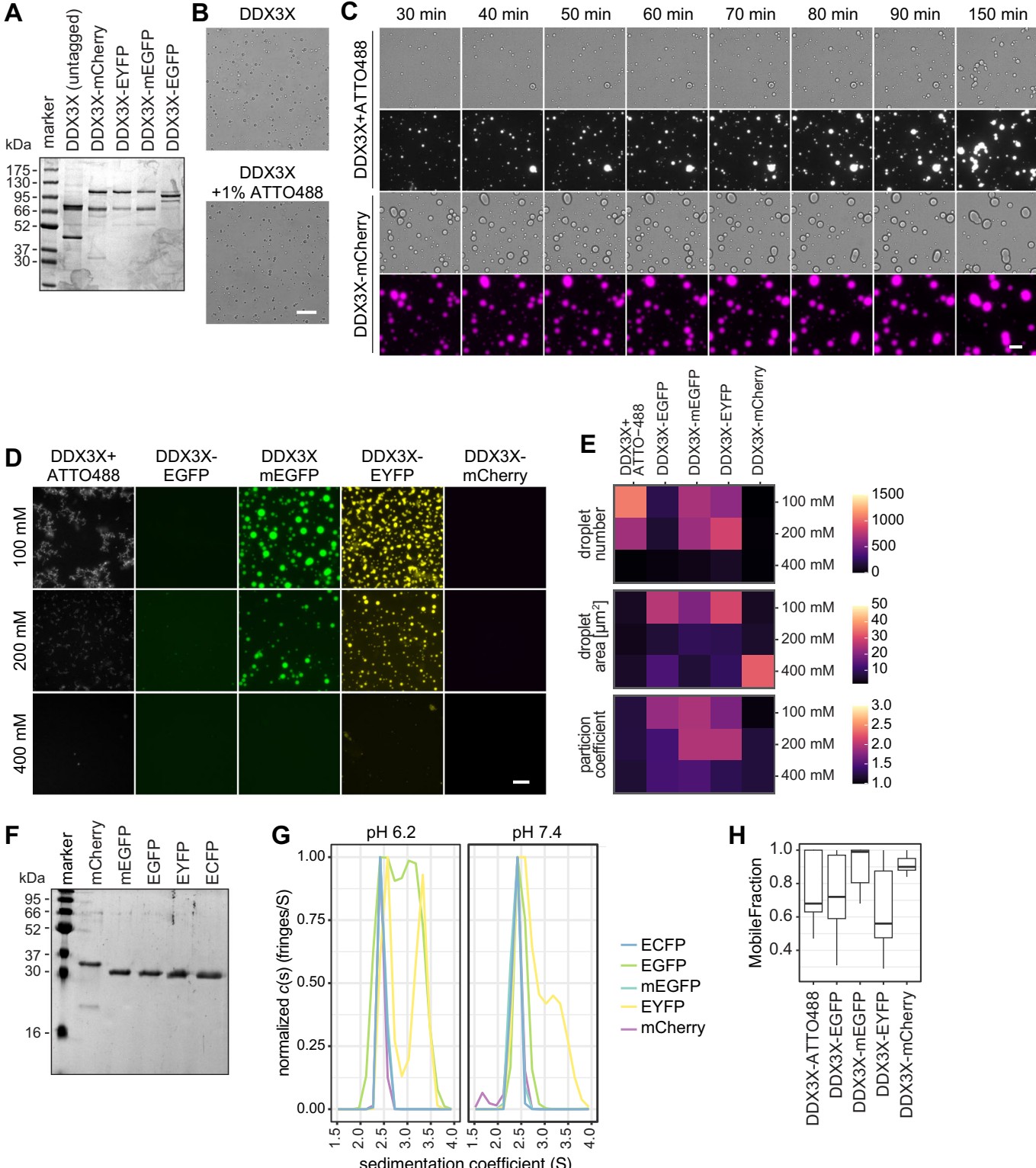

◀ **Figure EV2. Characterization of DDX3X condensates and biophysical properties of fluorescent protein tags.**

(A) Coomassie stained SDS-PAGE with 0.5 µg of the respective DDX3X protein per lane. (B) In vitro condensation assay with 5 µM untagged DDX3X or 5 µM untagged DDX3X + 1% ATTO-DDX3X spike-in in 25 mM sodium phosphate buffer at pH 6.2, 150 mM NaCl, 2 mM MgCl$_2$, 0.5 mg/mL BSA, 0.05 mg/mL poly(U), 0.5 mM ATP, 0.5 mM DTT. Incubated at 25 °C for 30 min before imaging. Scale bar: 20 µm. (C) Time series of in vitro condensation assay with 5 µM DDX3X + 1% ATTO488-DDX3X spike-in and DDX3X-mCherry in 25 mM sodium phosphate buffer at the pH 6.2, 150 mM NaCl, 2 mM MgCl$_2$, 0.5 mg/mL BSA, 0.05 mg/mL poly(U), 0.5 mM ATP, 0.05 mM DTT. Incubated at 25 °C for the indicated time before imaging. Scale bar: 20 µm. (D) In vitro condensation assay with 5 µM DDX3X (tagged or untagged + 1% ATTO488-DDX3X spike-in) at pH 7.4, 100, 200 or 400 mM NaCl, 2 mM MgCl$_2$, 0.5 mg/mL BSA, 0.05 mg/mL poly(U), 0.5 mM ATP, 0.5 mM DTT. Incubated at 25 °C for 30 min before imaging. Scale bar: 20 µm. (E) Quantification of in vitro condensation assay in (D): number of condensates per 0.1 mm$^2$, droplet area and PC, 4 well positions were imaged in three independent replicates. (F) Coomassie stained SDS-PAGE with 0.5 µg of the respective FP per lane. (G) Analytical ultracentrifugation of 0.5 mg/mL of the respective fluorescent protein in 25 mM phosphate buffer pH 6.2 or 7.4, 150 mM NaCl, 50 mM KCl, 1% glycerol, 1 mM MgCl$_2$ and 0.5 mM β-mercaptoethanol. Values were normalized to the highest value for each construct. (H) FRAP of in vitro DDX3X condensates of similar size. 5 µM DDX3X in 25 mM sodium phosphate buffer at pH 6.2, 100 mM NaCl, 2 mM MgCl$_2$, 0.5 mg/mL BSA, 0.05 mg/mL poly(U), 0.5 mM ATP, 0.5 mM DTT. Condensates were matured for 1 h at 25 °C before FRAP. Mobile fraction was analyzed. Box plots show the median (center line), 25th–75th percentiles (bounds of box), and whiskers=1.5x IQR. $N = 3$, n ≥ 23 condensates.

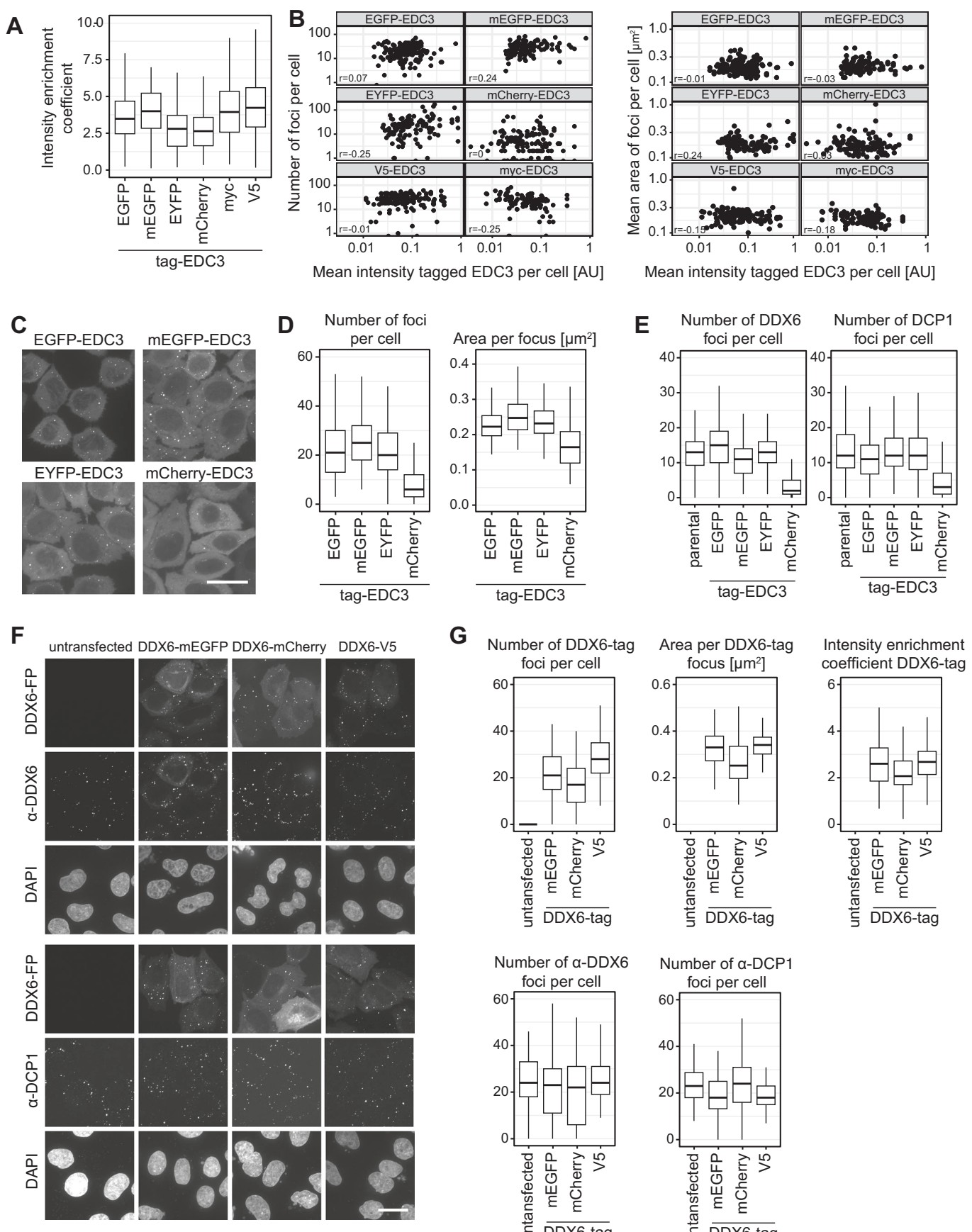

◀ **Figure EV3. EDC3 tag effects do not correlate with expression levels, and DDX6 is not influenced by tagging.**

Box plots show the median (center line), 25th–75th percentiles (bounds of box), and whiskers=1.5x IQR. (A, B) HeLa K cells were transiently transfected with EDC3 plasmids for 24 h, and stressed with 500 μM sodium arsenite for 30 min before fixation. V5-/myc-EDC3 were visualized by immunostaining with the respective antibodies. Quantification of EDC3 foci of cells in Fig. 3A: intensity enrichment coefficient (median intensity per focus/ median intensity of the cell) (B), number and area relative to the cellular expression level. Pearson correlation coefficient (r) is indicated. $N = 3$, $n \geq 110$. (C, D) Stable inducible HeLa cell lines were induced with doxycycline for 24 h to express the respective EDC3-tag constructs. Cells were stressed with 500 μM sodium arsenite for 30 min and imaged live. Scale bar: 20 μm. Quantification of number of EDC3 foci and their area (D). $N = 3$, $n \geq 100$ cells. (E) Stable inducible HeLa cell lines were induced with doxycycline for 24 to express the respective EDC3-tag constructs. Cells were stressed with 500 μM sodium arsenite for 30 min before fixation. Cells from the same well were either immunostained for DDX6 or DCP1 with respective antibodies. Scale bar: 20 μm. Quantification of number of DDX6 and DCP1 foci (I). $N = 3$, $n \geq 140$ cells. (F, G) HeLa K cells were transiently transfected with DDX6 plasmids for 24 h and stressed with 500 μM sodium arsenite for 30 min before fixation. Cells from the same well were either immunostained for DDX6 or DCP1 with respective antibodies. Scale bar: 20 μm. Quantification of number of DDX6 and DCP1 foci (G). $N = 3$, $n \geq 85$.

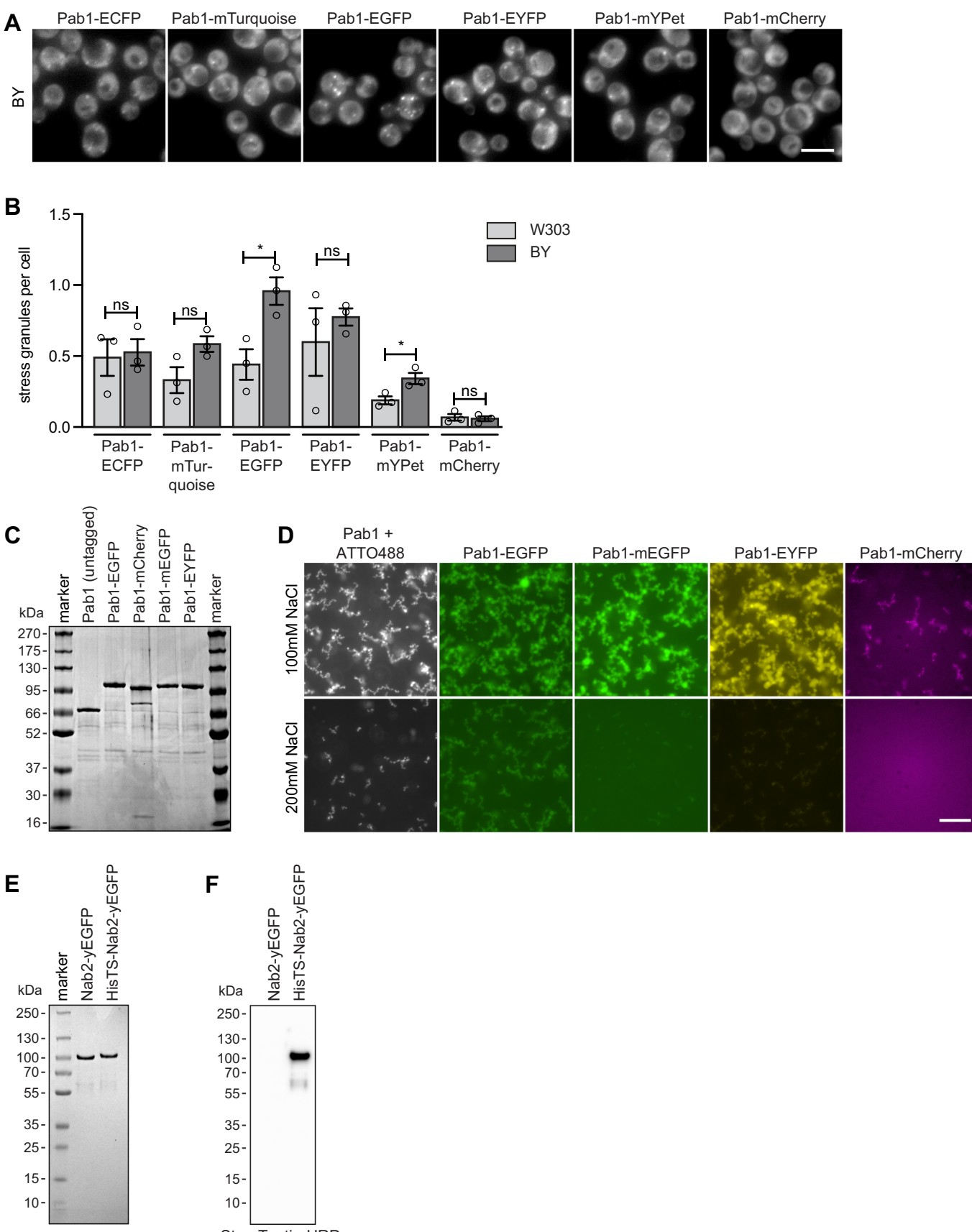

◀   **Figure EV4.   Pab1 tagging alters condensate formation in yeast and in vitro.**

(A) *S. cerevisiae* BY strains expressing Pab1 fused with the indicated FP tag were cultivated with 3% glycerol for 45 min prior to live imaging to induce SG formation. p(Pab1-EGFP) = 0.024, p(Pab1-mYPET)=0.034. Representative maximum Z-projections are shown. Scale bar: 5 µm. (B) Quantification of SG per cell in (A) and Fig. 4A, mean +/− SEM, $N = 3$, $n \geq 1500$ cells per strain. ns: non-significant, *$P \leq 0.05$ in unpaired *t* test W303 against BY. (C) Coomassie stained SDS-PAGE with 0.5 µg of the respective Pab1 protein per lane. (D) FP channel images corresponding to DIC images in Fig. 4C. Scale bar: 20 µm. (E) Coomassie stained SDS-PAGE with 0.5 µg of the respective Nab2 protein per lane. (F) Western Blot with 0.5 µg of the respective Nab2 protein per lane, stained with Strep Tactin conjugated with horse radish peroxidase (HRP).

                                                     

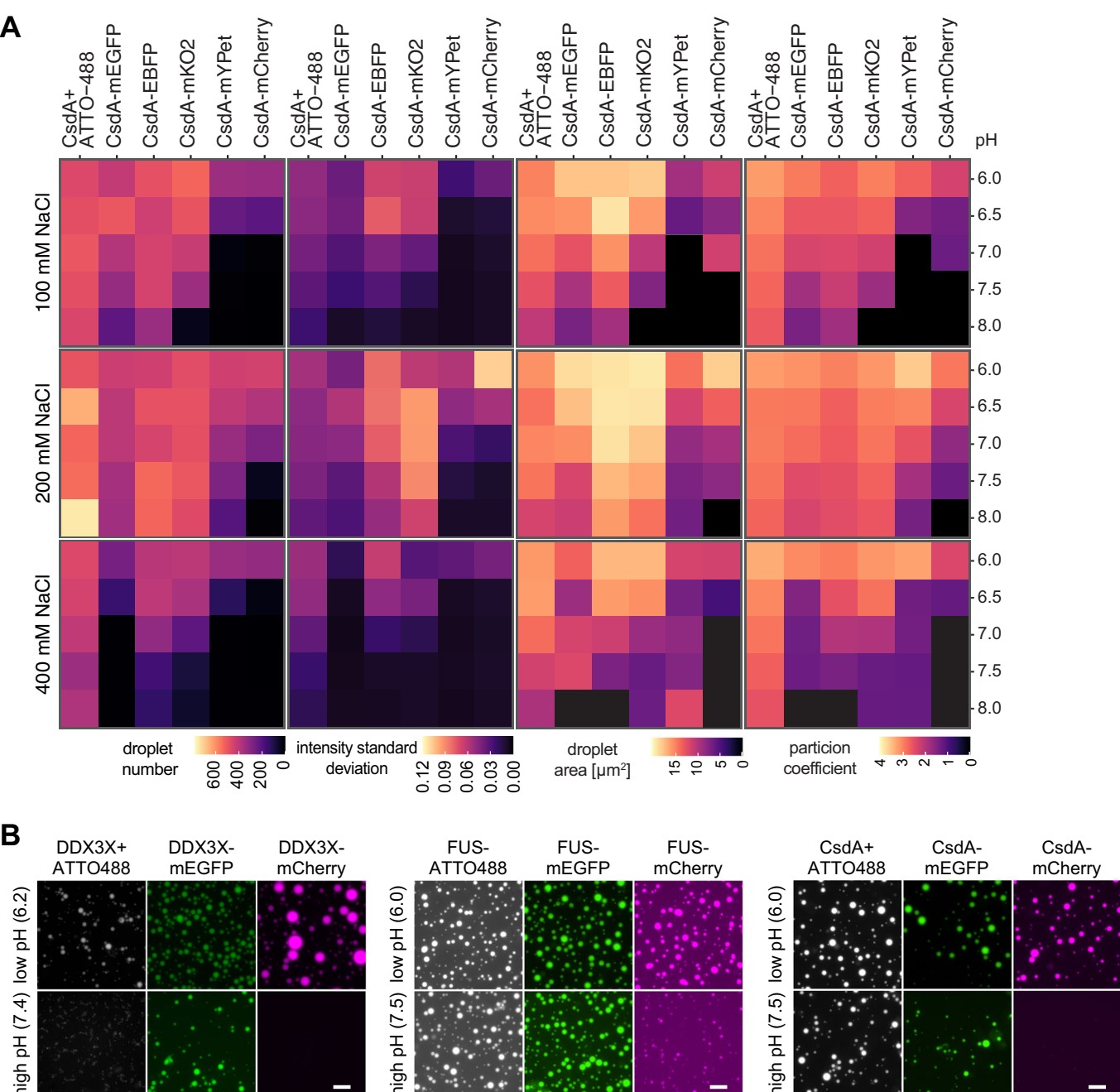

**Figure EV5. FP tags and physicochemical parameters modulate CsdA in vitro condensation.**

mEGFP and mCherry show distinct pH-dependent effects for DDX3X, FUS, and CsdA. (A) Quantification of in vitro condensation assay in (Appendix Fig. S6BA) number of droplets per 0.1 mm², droplet area, image intensity standard deviation and PC, N = 3. (B) Comparative assessment of DDX3X, FUS, and CsdA in vitro condensation when tagged with mEGFP or mCherry at 200 mM NaCl at pH 6.2/7.4 for DDX3X and pH 6.0/7.5 for FUS and CsdA; images are identical to those displayedin Fig. 2C (DDX3X), Fig. 5C (FUS), Appendix Fig. S6B (CsdA). Scale bar: 20 µm.

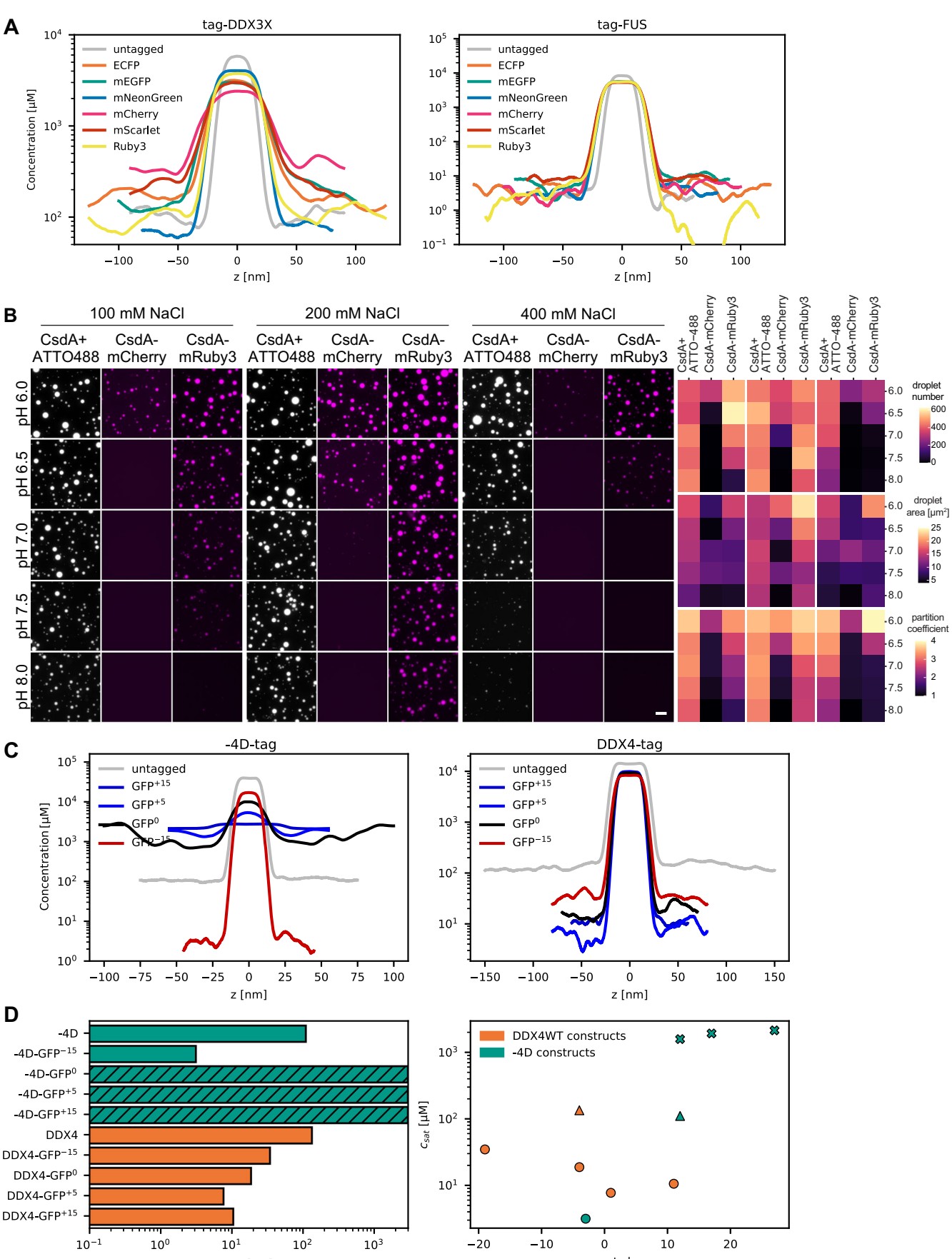

◄ **Figure EV6.   Phase coexistence simulations indicate that tag position and fluorescent protein charge determine condensation behavior of fusion proteins.**

Phase coexistence simulations were performed at 293 K with an ionic strength of 0.15 M using CALVADOS 3. (**A**) Equilibrium density profiles of six FPs N-terminally fused with full-length DDX3X (left) and full-length FUS (right). (**B**) In vitro condensation assay with 2 μM CsdA (tagged or untagged + 1% ATTO488-CsdA spike-in) in 25 mM sodium phosphate buffer at the indicated pHs and NaCl concentrations, 2 mM MgCl$_2$, 0.5 mg/mL BSA, 0.05 mg/mL poly(U). Incubated at 25 °C for 1 h before imaging. Representative images are shown. Scale bar: 20 μm. Quantification: number of droplets per 0.1 mm$^2$, droplet area and PC, $N = 3$. (**C**) Equilibrium density profiles of the fully intrinsically disordered proteins -4D (left) and truncated DDX4 (right) N-terminally fused with GFP variants with varying net charge. (**D**) Simulated saturation concentrations (c$_{sat}$, left) and correlation between c$_{sat}$ and net charge of for DDX4 (orange) or -4D (green) constructs with N-terminal tags. Circles represent tagged proteins, triangles untagged proteins. Hatched bars (left) and crosses (right): We did not see a stable condensed phase.

