## [Peer Review File · EMBO Reports]

Fluorescent protein and peptide tags alter condensate formation and dynamics in vivo and in vitro

Kerstin Dörner, Michelle Gut, Daan Overwijn, Fan Cao, Matej Siketanc, Stephanie Heinrich, Nicole Beuret, Justin Meyer, Timothy Sharpe, Kresten Lindorff-Larsen, and Maria Hondele

Corresponding author(s): Maria Hondele (maria.hondele@unibas.ch)

Review Timeline:

Submission Date:	3rd Mar 25
Editorial Decision:	22nd Apr 25
Revision Received:	22nd Jul 25
Editorial Decision:	9th Sep 25
Revision Received:	29th Sep 25
Accepted:	23rd Oct 25

Transaction Report:

Dear Dr. Hondele

Thank you for the submission of your research manuscript to our journal. We have now received the full set of referee reports that is copied below.

As you will see, the referees acknowledge that the findings are interesting and that the conclusions are overall supported by the data presented but they also raise a number of concerns and have suggestions how to further strengthen the data.

Given these constructive comments, we would like to invite you to revise your manuscript with the understanding that the referee concerns must be fully addressed and their suggestions taken on board. Please address all referee concerns in a complete point-by-point response. Acceptance of the manuscript will depend on a positive outcome of a second round of review. It is EMBO Reports policy to allow a single round of revision only and acceptance or rejection of the manuscript will therefore depend on the completeness of your responses included in the next, final version of the manuscript.

We realize that it is difficult to revise to a specific deadline. In the interest of protecting the conceptual advance provided by the work, we recommend a revision within 3 months (July 22nd). Please discuss the revision progress ahead of this time with the editor if you require more time to complete the revisions.

I am also happy to discuss the revision further via e-mail or a video call, if you wish.

=====
IMPORTANT NOTE:

We perform an initial quality control of all revised manuscripts before re-review. Your manuscript will FAIL this control and the handling will be delayed IN CASE the following APPLIES:

- 1) A data availability section providing access to data deposited in public databases is missing. If you have not deposited any data, please add a sentence to the data availability section that explains that.
- 2) Your manuscript contains statistics and error bars based on $n=2$. Please use scatter blots in these cases. No statistics should be calculated if $n=2$.

=====

- 1) a .docx formatted version of the manuscript text (including legends for main figures, EV figures and tables). Please make sure that the changes are highlighted to be clearly visible.
- 2) individual production quality figure files as .eps, .tif, .jpg (one file per figure). Please download our Figure Preparation Guidelines (figure preparation pdf) from our Author Guidelines pages <https://www.embopress.org/page/journal/14693178/authorguide> for more info on how to prepare your figures.
- 3) a .docx formatted letter INCLUDING the reviewers' reports and your detailed point-by-point responses to their comments. As part of the EMBO Press transparent editorial process, the point-by-point response is part of the Review Process File (RPF), which will be published alongside your paper.
- 4) a complete author checklist, which you can download from our author guidelines (<<https://www.embopress.org/page/journal/14693178/authorguide>>). Please insert information in the checklist that is also reflected in the manuscript. The completed author checklist will also be part of the RPF.

5) Please note that all corresponding authors are required to supply an ORCID ID for their name upon submission of a revised manuscript (<<https://orcid.org/>>). Please find instructions on how to link your ORCID ID to your account in our manuscript tracking system in our Author guidelines

(<<https://www.embopress.org/page/journal/14693178/authorguide#authorshipguidelines>>)

6) We replaced Supplementary Information with Expanded View (EV) Figures and Tables that are collapsible/expandable online. A maximum of 5 EV Figures can be typeset. EV Figures should be cited as 'Figure EV1, Figure EV2' etc... in the text and their respective legends should be included in the main text after the legends of regular figures.

7) Please note that a Data Availability section at the end of Materials and Methods is now mandatory. In case you have no data that requires deposition in a public database, please state so instead of refereeing to the database.

See also < <https://www.embopress.org/page/journal/14693178/authorguide#dataavailability>>. Please note that the Data Availability Section is restricted to new primary data that are part of this study.

Additional information on source data and instruction on how to label the files are available

<<https://www.embopress.org/page/journal/14693178/authorguide#sourcedata>>

10) Figure legends and data quantification:

- the name of the statistical test used to generate error bars and P values,
- the EXACT p-values,
- the number (n) of independent experiments (please specify technical or biological replicates) underlying each data point,
- the nature of the bars and error bars (s.d., s.e.m.)

- If the data are obtained from n {less than or equal to} 5, show the individual data points in addition to the SD or SEM.

- If the data are obtained from n {less than or equal to} 2, use scatter blots showing the individual data points.

11) Our journal encourages inclusion of *data citations in the reference list* to directly cite datasets that were re-used and obtained from public databases. Data citations in the article text are distinct from normal bibliographical citations and should directly link to the database records from which the data can be accessed. In the main text, data citations are formatted as follows: "Data ref: Smith et al, 2001" or "Data ref: NCBI Sequence Read Archive PRJNA342805, 2017". In the Reference list, data citations must be labeled with "[DATASET]". A data reference must provide the database name, accession number/identifiers and a resolvable link to the landing page from which the data can be accessed at the end of the reference. Further instructions are available at <<https://www.embopress.org/page/journal/14693178/authorguide#referencesformat>>.

12) All Materials and Methods need to be described in the main text using our 'Structured Methods' format. According to this format, the Methods section includes a Reagents and Tools Table (listing key reagents, experimental models, software and relevant equipment and including their sources and relevant identifiers) followed by a Methods and Protocols section describing

the methods, ideally using a step-by-step protocol format. The aim is to facilitate adoption of the methodologies across labs. Please download and fill our Reagents and Tools Table template (.docx), which you can find in our author guidelines: <https://www.embopress.org/page/journal/14693178/authorguide#structuredmethods>.

An example of a Method paper with Structured Methods can be found here: <https://www.embopress.org/doi/10.15252/msb.20178071>.

13) As part of the EMBO publication's Transparent Editorial Process, EMBO Reports publishes online a Review Process File to accompany accepted manuscripts. This File will be published in conjunction with your paper and will include the referee reports, your point-by-point response and all pertinent correspondence relating to the manuscript.

Yours sincerely,

=====

Referee #1:

This research article emphasizes an important area of concern for the fields of cell biology and condensate biology, specifically that affinity tags or fluorescent protein (FP) tags can influence the behavior of proteins known to either phase separate in vitro or form puncta in cells. Similar studies are coming out with similar findings (<https://www.molbiolcell.org/doi/10.1091/mbc.E24-01-0013>, <https://doi.org/10.1091/mbc.E24-11-0521>, [https://www.cell.com/biophysj/fulltext/S0006-3495\(23\)04117-6](https://www.cell.com/biophysj/fulltext/S0006-3495(23)04117-6), among others). Similarly, fixation artifacts can also influence condensate properties (<https://elifesciences.org/articles/85671>). Therefore, this manuscript highlights a timely issue given many condensate-based studies use tags to image protein subcellular localization. The authors collect extensive imaging data on several proteins in vitro (purified protein-based studies) and in cells (FP-tagged or affinity (HA/Myc/ALFA, etc.)-tagged), and the authors systematically examine their properties including number of puncta, size of puncta, and protein enrichment inside puncta. Not only do the authors examine these proteins in mammalian cell lines, but they also examine puncta formation in yeast. While the authors primarily focus on transient expression studies (common), they also selectively examine stable cell lines with inducible promoter-controlled expression of proteins. Importantly, they find that the FP tag itself can affect condensate properties and, in some cases, significantly reduce puncta formation (e.g., mCherry). The authors correlate some of these in-cell behaviors with what they observe from in-vitro condensate experiments. Furthermore, the effects are not always predictable as in some proteins are more affected than others (e.g., G3BP puncta formation under stress to form stress granules does not appear to be generally affected by the type of tag). As expected, various conditions impact their observations in cells - pH, salt, etc. Tags can also influence RNA-binding properties as demonstrate for Nab2. Using molecular dynamics simulations, the authors also find that the overall charge of the fluorescent protein may impact condensate formation in a predictable way for one representative protein. Overall, the study presents comprehensive results that should be widely read by the community. Important considerations are offered including the need to examine several FP tags when conducting in-cell assays, and further complementing in-cell work with in-vitro studies particularly to assess whether the tag influences protein behavior in cells. The overall message is that tags can influence behavior of proteins and that it is context-dependent. Therefore, there is not a single solution to overcoming this problem. Below I have written additional recommendations to further improve this important manuscript.

1) The authors' results may be further impacted by the location of the tag on the protein (N-terminus vs. C-terminus vs. internal). The decision on where to place the tag is an important contributor to effects on phase separation, particularly if the tag is next to a functional domain. In the last figure (6), the authors test the effects of positioning the tag computationally but do not elaborate on this point in the text (unless I missed this). There is a very brief mention of this within the second paragraph of the discussion,

but should be expanded upon.

2) In the opening results' section, the authors mention that, for ECFP and EYFP, the granules became "non-spherical, indicating protein aggregation". I would be careful with this statement given that the appearance could be a consequence of just large puncta perhaps restricted in shape due to adjoining organelles or other cellular assemblies. Without complementary FRAP experiments, it is difficult to assess the state of these 'granules'. Interestingly, the FRAP data for in-vitro DDX3X suggests that the large amorphous droplets of DDX3X-mCherry are the most liquid-like of the set of tested DDX3X-FP constructs.

3) FPs, especially from the red wavelength spectrum, decrease the 'intensity enrichment coefficient'. Why is that? Can the authors expand on this point?

4) In the section regarding 'FP tags modulate in vitro condensation of E. coli CsdA...' the authors mention that "CsdA-EBFP2 was closest to untagged+ATTO CsdA". This is unclear - what are the metrics for supporting this statement and could this be quantified for the reader?

5) On pg. 11 the authors mention GFP multivalency for Figure 6B/C, ED Fig 6B/C. This is not made clear in the figures and I presume this is discussing the tendency for certain FPs to form dimers? Please elaborate.

6) Regarding net charge of FPs and in silico phase separation behavior: In Figure 6, -4D exhibits a charge-dependent effect on saturation concentration. However, the Extended Data figure 6 are incorrect in panel C. The colors between the left and right panels do not match. Additionally, the data points show in the right-most plot seem to be identical to those shown in Figure 6C-right, but with one data point missing. To further improve the data, could the authors perform additional simulations with GFPs of +10 and -10 overall net charge? Additionally, other studies have suggested that charge effects can influence condensate-making properties such as [https://www.cell.com/biophysj/fulltext/S0006-3495\(23\)04117-6](https://www.cell.com/biophysj/fulltext/S0006-3495(23)04117-6). Overall, the point about FP tag overall charge is an important one, and I wish this would be further complemented by experimental data using these different GFP variants.

7) Have the authors tried to correlate some of their in-cell observations with properties including net charge of the FPs, and/or whether FP placement on the tested proteins is proximal to a portion of the protein that is important for mediating protein-protein interactions / folded domain /intrinsically-disordered region?

8) Would also like to note that supplementary tables were missing from the PDF for this manuscript. I did my best to find supplementary tables from the biorxiv version of the manuscript although the supplementary table numbering does not match.

Referee #2:

In this study, Dörner et al. investigated how fluorescent protein tags influence the morphology and dynamics of biomolecular condensates both in cells and in vitro. Their key finding stems from a systematic immunofluorescence screen of 16 C-terminally fused fluorescent proteins on DDX3X following stress induced by arsenite treatment. Overexpression of these tagged constructs for 24 hours, followed by fixation and imaging, revealed that GFP-derived tags tend to promote condensate formation and potentially aggregation -- evidenced by misshapen condensates -- whereas red-derived tags (and in particular mCherry) had the opposite effect. These observations were supported by in vitro assays and client-scaffold experiments in cells, suggesting the net and surface charge of the fluorescent proteins as key factors. This is a carefully executed and important study given the widespread use of fluorescent fusion proteins in cell biology, and advances our understanding of how tagging can affect protein behaviour in crowded environments. I have one major comment and a few minor suggestions.

Major comment

1. In this study, the workhouse assay combined overexpression (either > 1:1 or 1:1 with endogenous levels) for 24 hours, followed by 30 minutes arsenite treatment and then fixation and imaging. Biomolecular condensates are complex physicochemical environments and their formation is driven by weak multivalent interactions. There are difficulties with fixing condensates both in cells and in vitro (antibody penetration can be a major problem, amongst other issues). It is likely beyond scope to repeat all experiments with live imaging. But, as this study uses fluorescent proteins, could the authors perform representative live-imaging-based examples that demonstrate that the fixation is not compromising their main points? Also, a discussion on fixation and how this plays a role in disrupting condensate systems would be helpful and informative.

Minor comments

1. Can you include a reference where it is shown that DCP1 is a good marker for PBs? What is your metric for a good marker? Perfectly overlapping to diffraction limit?
2. Can you provide an explanation and include in text why the mCherry-tagged and the G3BP1 localization is different?
3. Can you give more background on the Calvados system when it is introduced?

Referee #3:

In the manuscript "Fluorescent protein and peptide tags alter condensate formation and dynamics in vivo and in vitro" the authors test the effect of a range of fluorescent tags on DDX3X, Pab1, EDC3, CsdA, FUS and Nab2 condensate forming proteins. This work is a thorough effort to map the effect of commonly used protein tags on a number of often studied condensate forming proteins. While effects are often quite target-protein specific, the authors do provide actionable insights and provide useful recommendations for experimental design. The study is well designed, executed and represented. A few other studies have recently addressed the same tagging concern (Barkley et al., 2024; Pandey et al., 2024; Fatti, Khawaja and Weis, 2025), but all focus on a single or small selection of proteins of interest. Therefore, this work represents the first systematic effort to catalogue and understand tag-effects on biomolecular condensates and will provide an important resource for the condensate biology field.

I have only a few minor comments and suggestions for improvement that the authors may wish to address.

Fig 1A. The effects of the different tags are tested using transient over-expression, which leads to highly variable expression levels cell to cell, all the way from no expression to several-fold over endogenous levels. This is evidenced by the variability displayed in the different quantification methods. How reliable is the interpretation of the different tag-effects under such high variability? A more controlled experiment would be endogenous labeling of DDX3X with the different labels, but I appreciated this is a significant effort. At least the authors should compare the variability of the transiently expressed, tagged versions with that of the IF detected endogenous DDX3X, to evaluate the level of variability that is introduced by transient over-expression.

The authors find an interesting lack of correlation between the level of over-expression and the propensity to form condensates that flies in the face of LLPS mechanics but is more widely observed in the condensate field. Can the authors comment on why there is no strong correlation between the level of protein and condensate formation?

Do the exogenously expressed tagged proteins also affect endogenous condensates? Can the authors perform an IF for DDX3X together with the labeled proteins and see if 1) the exogenous protein partitions into endogenous condensates and 2) if the effects of the specific tags also apply to endogenous DDX3X condensates? Currently only G3BP1 is used as a marker.

Fig 3C/D shows that tag-EDC3 affects DCP1 in P-bodies. Is the condensate-perturbing effect also POI dependent?

While it's important to know the effect of tags on condensate formation, a typical experimental setup compares two variants of the same protein (e.g. wt v.s. disease mutant) both tagged with the same fluorophore. In this case the tag-effects are internally controlled since both proteins carry the same tag. Can the authors comment on or experimentally address the risk of tags unduly affecting such a comparison?

Fig 4D. The authors claim HisTS alters RNA specificity but show no evidence for that. Instead an increased condensate propensity across the board is observed.

References

- Barkley, R.J.R. et al. (2024) "Fluorescent protein tags affect the condensation properties of a phase-separating viral protein.," *Molecular Biology of the Cell*, 35(7), p. ar100. doi:10.1091/mbc.E24-01-0013.
- Fatti, E., Khawaja, S. and Weis, K. (2025) "The dark side of fluorescent protein tagging-the impact of protein tags on biomolecular condensation.," *Molecular Biology of the Cell*, 36(3), p. br10. doi:10.1091/mbc.E24-11-0521.
- Pandey, N.K. et al. (2024) "Fluorescent protein tagging promotes phase separation and alters the aggregation pathway of huntingtin exon-1.," *The Journal of Biological Chemistry*, 300(1), p. 105585. doi:10.1016/j.jbc.2023.105585.

Dörner et al., Response to Reviewer's Comments:

Ref: Submission ID: EMBOR-2025-61459V2

We thank the reviewers for their positive assessment of our work. Their thoughtful comments and constructive suggestions have significantly helped us to clarify, refine, and improve the manuscript. In the revised version of our manuscript, we have implemented several key improvements:

1. Expanded computational analysis of tag charge effects:

We performed additional *in silico* simulations using an extended set of GFP charge variants, strengthening the correlation between tag charge and phase separation propensity (Fig. 6 B/C).

2. New live-cell imaging to control for fixation artifacts:

To address concerns about fixation-dependent artifacts, we conducted live-cell experiments for DDX3X and EDC3 in stable cell lines. The results closely mirrored those from fixed cells, supporting the robustness of our observations (EV Fig. 1C/D and EV Fig. 3C/D).

3. mRuby3 as potential FP tag choice in the red wavelength spectrum:

Since *in silico* simulations and experiments in cells with DDX3X indicated that mRuby3 might be a FP tag that interferes relatively little with condensation, we performed additional experiments with recombinant CsdA-mRuby3 *in vitro*. We could verify that compared to other red FPs, in particular mCherry, mRuby3 seems to have less influence on condensation *in vitro*.

4. Staining for endogenous DDX3X in tagged lines:

We performed new co-staining experiments with an anti-DDX3X antibody to assess the impact of exogenously expressed FP-tagged proteins on endogenous DDX3X condensates. These experiments show that the observed effects are specific to the tagged construct and do not broadly disrupt behavior of endogenous DDX3X (EV Fig. 1E/F, Appendix Fig. 3F).

5. DDX6 as additional PB maker:

We added co-staining experiments for the PB marker DDX6 in cells expressing FP-EDC3 fusion proteins. We find that expression of mCherry-EDC3 can impair recruitment of other endogenous PB components, as judged by DCP1 and DDX6 immunofluorescence. In addition, we performed experiments with C-terminally FP-tagged DDX6 and co-staining with the DCP1 antibody, where we find that expression of FP-DDX6 has little influence on PB appearance (EV Fig. 3E–G, Appendix Fig. 4C/D).

6. Clarified figures, corrected errors, and added references

Please find below our point-by-point responses to the reviewers' comments in blue, with sections of newly added text highlighted in *violet*.

Referee #1:

This research article emphasizes an important area of concern for the fields of cell biology and condensate biology, specifically that affinity tags or fluorescent protein (FP) tags can influence the behavior of proteins known to either phase separate in vitro or form puncta in cells. Similar studies are coming out with similar findings (<https://www.molbiolcell.org/doi/10.1091/mbc.E24-01-0013>, <https://doi.org/10.1091/mbc.E24-11-0521>, [https://www.cell.com/biophysj/fulltext/S0006-3495\(23\)04117-6](https://www.cell.com/biophysj/fulltext/S0006-3495(23)04117-6), among others). Similarly, fixation artifacts can also influence condensate properties (<https://elifesciences.org/articles/85671>). Therefore, this manuscript highlights a timely issue given many condensate-based studies use tags to image protein subcellular localization.

The authors collect extensive imaging data on several proteins in vitro (purified protein-based studies) and in cells (FP-tagged or affinity (HA/Myc/ALFA, etc.)-tagged), and the authors systematically examine their properties including number of puncta, size of puncta, and protein enrichment inside puncta. Not only do the authors examine these proteins in mammalian cell lines, but they also examine puncta formation in yeast. While the authors primarily focus on transient expression studies (common), they also selectively examine stable cell lines with inducible promoter-controlled expression of proteins. Importantly, they find that the FP tag itself can affect condensate properties and, in some cases, significantly reduce puncta formation (e.g., mCherry). The authors correlate some of these in-cell behaviors with what they observe from in-vitro condensate experiments. Furthermore, the effects are not always predictable as in some proteins are more affected than others (e.g., G3BP puncta formation under stress to form stress granules does not appear to be generally affected by the type of tag). As expected, various conditions impact their observations in cells - pH, salt, etc. Tags can also influence RNA-binding properties as demonstrate for Nab2. Using molecular dynamics simulations, the authors also find that the overall charge of the fluorescent protein may impact condensate formation in a predictable way for one representative protein.

Overall, the study presents comprehensive results that should be widely read by the community. Important considerations are offered including the need to examine several FP tags when conducting in-cell assays, and further complementing in-cell work with in-vitro studies particularly to assess whether the tag influences protein behavior in cells. The overall message is that tags can influence behavior of proteins and that it is context-dependent. Therefore, there is not a single solution to overcoming this problem. Below I have written additional recommendations to further improve this important manuscript.

We thank the reviewer for the positive and encouraging assessment of our work. We are pleased they recognize the timeliness and broad relevance of our comprehensive, multi-system study for the condensate field. We agree that it is key to uncovering both general trends and context-dependent effects of tagging that are not apparent from isolated case studies. Your insightful comments clearly helped to improve our manuscript and made it easier to read.

1) The authors' results may be further impacted by the location of the tag on the protein (N-terminus vs. C-terminus vs. internal). The decision on where to place the tag is an important contributor to effects on phase separation, particularly if the tag is next to a functional domain. In the last figure (6), the authors test the effects of positioning the tag computationally but do

not elaborate on this point in the text (unless I missed this). There is a very brief mention of this within the second paragraph of the discussion, but should be expanded upon.

We agree with the reviewer that the position of the tag is critical for both cell biological and biochemical experiments. To emphasize this point, we expanded the statement in the results section and added two sentences to the discussion section.

“While FP fusions at both the N- and C-termini of FUS generally exhibited similar condensation behavior, DDX3X showed noticeable differences depending on the tagging location. For instance, when ECFP, mEGFP, mCherry, or mScarlet are fused to the C-terminus of DDX3X, the simulations do not show stable condensation. In contrast, fusing the same FPs to the N-terminus results in more robust phase separation. These findings suggest that the distinct phase behavior of DDX3X and FUS upon N- or C-terminal tagging arises from local differences in molecular properties near the termini, such as amino acid composition, charge distribution, and domain architecture.”

“Differences likely arise from alterations in the molecular surface properties of the tagged POI — such as surface charge or hydrophobicity — as well as from the tag’s position relative to condensation-mediating residues (see Appendix Fig. 8). Consequently, tagging at the N- versus C-terminus can lead to markedly different outcomes. Obviously, such effects are not unique to condensate-forming proteins and have been documented for other protein classes as well^{45,50} .”

2) In the opening results' section, the authors mention that, for ECFP and EYFP, the granules became "non-spherical, indicating protein aggregation". I would be careful with this statement given that the appearance could be a consequence of just large puncta perhaps restricted in shape due to adjoining organelles or other cellular assemblies. Without complementary FRAP experiments, it is difficult to assess the state of these 'granules'. Interestingly, the FRAP data for in-vitro DDX3X suggests that the large amorphous droplets of DDX3X-mCherry are the most liquid-like of the set of tested DDX3X-FP constructs.

We agree with the reviewer that several molecular mechanisms could cause the observed change in appearance. We changed the text as follows:

“For ECFP and EYFP the granules became non-spherical, which could suggest protein aggregation or shape restriction by other cellular structures.”

We also tuned down wording in several other locations.

3) FPs, especially from the red wavelength spectrum, decrease the 'intensity enrichment coefficient'. Why is that? Can the authors expand on this point?

Thank you for this thoughtful comment. We believe that the similar behavior observed among the 'green' proteins is at least partly due to their shared evolutionary origin (see Appendix Fig. 2). In contrast, the 'red' proteins originate from different lineages - both to GFP and amongst themselves - and display more divergent properties.

For example, mCherry and mScarlet-I tend to suppress condensation at physiological pH, whereas mRuby3 appears to be less disruptive both in cells (see Fig. 1A/B) and *in vitro* (see new data in EV Fig. 6B). To clarify this point, we added the following sentence to the manuscript:

“The divergent behavior of red and green fluorescent proteins likely reflects differences in their evolutionary origins rather than their emission wavelength, as many green variants are derived from GFP and thus likely share similar biophysical properties (Appendix Fig. 2).”

EV Fig 6:

(B) *In vitro* condensation assay with 2 μ M CsdA (tagged or untagged + 1% ATTO488-CsdA spike-in) in 25 mM sodium phosphate buffer at the indicated pHs and NaCl concentrations, 2 mM MgCl₂, 0.5 mg/mL BSA, 0.05 mg/mL poly(U). Incubated at 25°C for 1 h before imaging. Representative images are shown. Scale bar: 20 μ m. Quantification: number of droplets per 0.1 mm², droplet area, image intensity standard deviation and PC, N = 3.

4) In the section regarding 'FP tags modulate in vitro condensation of E. coli CsdA...' the authors mention that "CsdA-EBFP2 was closest to untagged+ATTO CsdA". This is unclear - what are the metrics for supporting this statement and could this be quantified for the reader?

Thank you for this comment, we agree that this statement is not sufficiently supported. We removed the sentence.

5) On pg. 11 the authors mention GFP multivalency for Figure 6B/C, ED Fig 6B/C. This is not made clear in the figures and I presume this is discussing the tendency for certain FPs to form dimers? Please elaborate.

Thank you for pointing this out. You are correct, we were originally referring to the known tendency of certain fluorescent proteins (FPs) to dimerize. However, during expansion of the dataset to include a wider range of GFP charge variants, we realized that FP dimerization does not substantially impact the modeling results. As a result, we revised the text and removed the reference to GFP dimerization for clarity.

6) Regarding net charge of FPs and in silico phase separation behavior: In Figure 6, -4D exhibits a charge-dependent effect on saturation concentration.

— However, the Extended Data figure 6 are incorrect in panel C. The colors between the left and right panels do not match. Additionally, the data points show in the right-most plot seem to be identical to those shown in Figure 6C-right, but with one data point missing.

Thank you very much for pointing out this error! We replaced the figure with an updated version containing the correct data (see below).

— To further improve the data, could the authors perform additional simulations with GFPs of +10 and -10 overall net charge? Additionally, other studies have suggested that charge effects can influence condensate-making properties such as [https://www.cell.com/biophysj/fulltext/S0006-3495\(23\)04117-6](https://www.cell.com/biophysj/fulltext/S0006-3495(23)04117-6).

Thank you for this thoughtful suggestion. We agree that the effect of charge is a particularly interesting aspect. We have added the suggested citation to our discussion.

To improve our dataset, we performed additional simulations using a broader range GFP charge variants, GFP⁻²⁰, GFP⁻¹⁵, GFP⁻¹⁰, GFP⁻⁵, GFP⁰, GFP⁺⁵, GFP⁺¹⁰, GFP⁺¹⁵, and GFP⁺²⁰. These new results reveal that for the -4D variant, the condensation promoting effect of GFP correlates with its overall charge. In contrast, for DDX4, the relationship is more complex (see new main Figure 6B/C).

Fig. 6 B/C:

(B) Equilibrium density profiles of the fully intrinsically disordered proteins -4D (left) and truncated DDX4 (right) N-terminally fused with GFP variants with varying net charge based with non-fluorescent GFPS65G, Y66G as GFP0.

(C) Simulated saturation concentrations (csat, left) and correlation between csat and net charge of for DDX4 (orange) or -4D (green) constructs with N-terminal tags. Hatched bars (left) and crosses (right) indicate systems that do not form stable condensates, and the values reported should be taken to mean that the csat values are high but uncertain. In the right panel, circles represent tagged proteins, and the two triangles represent the untagged IDRs of DDX4 and the -4D variant of A1 LCD.

We changed the text to read as follows:

“We shifted our focus to simpler systems consisting solely of IDRs, specifically the sequences -4D⁴¹ from the prion-like low-complexity domain of human hnRNPA1 (four native Asp residues substituted with Gly/Ser; net charge +12), and the isolated IDR of DDX4⁴³ (net charge -4), each fused to differently charged GFP variants (GFP⁻²⁰, GFP⁻¹⁵, GFP⁻¹⁰, GFP⁻⁵, GFP⁰, GFP⁺⁵,

GFP⁺¹⁰, GFP⁺¹⁵ and GFP⁺²⁰, superscript numbers indicate net charge). We note that even if GFP⁰ has a formal net charge of zero, it contains both positively and negatively charged residues. In line with previous findings on the recombinant protein⁴¹, untagged -4D undergoes phase separation during the simulations. Fusion with GFP reduces in silico condensation, an effect that is further exacerbated by increasingly positive GFP variants, likely due to electrostatic repulsion of the now highly positively charged fusion proteins (Fig. 6B, EV Fig. 6C, Appendix Fig. 11, Appendix Fig. 9/10/12). In contrast, attaching negatively charged GFP variants is predicted to significantly promote -4D phase separation, presumably by a combined effect of diminishing the net charge of the fusion protein and favorable interactions between GFP and the disordered region of -4D (Appendix Fig. 11/12). Similarly, untagged DDX4 IDR is also predicted to undergo phase separation, in agreement with data for the recombinant protein⁴³. The attachment of most GFP charge variants enhanced condensation, presumably facilitated by additional GFP-IDR interactions (Fig. 6B/C, EV Fig. 6C/D, Appendix Table S3). As for -4D, we observe that a protein with a net charge close to zero has the strongest driving force for phase separation, but that DDX4 IDR generally showed smaller effects. Interestingly, we find that for -4D the driving force becomes stronger as the net charge is titrated below zero, presumably reflecting additional attractive charge-charge interactions that overcome the global net charge effects. Together, these results demonstrate that net and surface charges of the FP and the POI can dominate the extent of condensation modulation, though other effects such as the number of charged residues also play a role.”

Overall, the point about FP tag overall charge is an important one, and I wish this would be further complemented by experimental data using these different GFP variants.

We fully agree with the reviewer that this is a very interesting point. To address the suggestion experimentally, we ordered the DNA sequences for the two GFP variants originally used in the simulations (GFP⁻¹⁵ and GFP⁺¹⁵) and cloned them into a vector expressing the -4D protein.

In the original publication demonstrating that -4D forms condensates (Bremer *et al.*), the protein was purified from inclusion bodies using a denaturing protocol, a method that typically works well for IDPs like -4D. However, because of the addition of the globular GFP tags, we preferred to deviate from the denaturing protocol and instead employed non-denaturing purification. We successfully purified the resulting fusion proteins (soluble up to 400 μ M in our storage buffer).

However, despite testing a wide range of conditions, we were unable to reproducibly induce condensation. Instead, both constructs tended to aggregate over time (see reviewer figure below).

Due to the three-month revision timeline, we were unfortunately unable to re-purify, or to reclone -4D into a different expression vector, or to reclone and purify CsdA as an alternative to -4D with the two GFP variants.

Reviewer figure 1: *in vitro* condensation attempts for the -4D-GFP constructs

7) Have the authors tried to correlate some of their in-cell observations with properties including net charge of the FPs, and/or whether FP placement on the tested proteins is proximal to a portion of the protein that is important for mediating protein-protein interactions / folded domain /intrinsically-disordered region?

Thank you for this thoughtful comment. We did not observe a clear correlation between the net charge of the FPs and their effects on condensation behavior (see Appendix Table S1), suggesting that additional factors are at play.

It is challenging to generalize the influence of domain architecture across all proteins tested, as they differ significantly both in domain structure and the regions responsible for phase separation. For DDX3X, the LCD1 region appears critical for LLPS (see new Appendix Figure 8 and <https://pubmed.ncbi.nlm.nih.gov/27180681/>, Fig. 7), we attach the tags to the C-terminus, and still DDX3X condensation is strongly influenced.

For G3BP1, LLPS seems to be facilitated by several domains, with the N-terminal dimerization domain playing an important role for oligomerization and LLPS (see PMIDs [32302572](https://pubmed.ncbi.nlm.nih.gov/32302572/), [32302570](https://pubmed.ncbi.nlm.nih.gov/32302570/), [32302571](https://pubmed.ncbi.nlm.nih.gov/32302571/)). In our study, we tagged G3BP1 at the N-terminus, but do not observe strong effects in our assays.

In conclusion, from the proteins we tested, we cannot make general statements about tag proximity, but agree with the reviewer that this might

be an important, POI-specific point. To allow the reader to orient themselves, we have created a schematic overview of all proteins tested and added a) the folded domains tested and b) the FuzDrop prediction scores (<https://fuzdrop.bio.unipd.it/predictor>) plotted across the sequence (Appendix figure 8).

8) Would also like to note that Appendix tables were missing from the PDF for this manuscript. I did my best to find Appendix tables from the biorxiv version of the manuscript although the Appendix table numbering does not match.

We are sorry to hear that the Appendix tables were not accessible. We are surprised to hear this, since we had uploaded them with the initial submission. We hope the reviewer can now access them without problem.

Referee #2:

In this study, Dörner et al. investigated how fluorescent protein tags influence the morphology and dynamics of biomolecular condensates both in cells and in vitro. Their key finding stems from a systematic immunofluorescence screen of 16 C-terminally fused fluorescent proteins on DDX3X following stress induced by arsenite treatment. Overexpression of these tagged constructs for 24 hours, followed by fixation and imaging, revealed that GFP-derived tags tend to promote condensate formation and potentially aggregation -- evidenced by misshapen condensates -- whereas red-derived tags (and in particular mCherry) had the opposite effect. These observations were supported by in vitro assays and client-scaffold experiments in cells, suggesting the net and surface charge of the fluorescent proteins as key factors.

This is a carefully executed and important study given the widespread use of fluorescent fusion proteins in cell biology, and advances our understanding of how tagging can affect protein behaviour in crowded environments. I have one major comment and a few minor suggestions.

We thank this reviewer for their positive feedback, and are grateful that they recognize the value of our careful and systematic study. We performed several experiments to address their very helpful comments, which in our opinion greatly strengthen the manuscript.

1. In this study, the workhouse assay combined overexpression (either > 1:1 or 1:1 with endogenous levels) for 24 hours, followed by 30 minutes arsenite treatment and then fixation and imaging. Biomolecular condensates are complex physicochemical environments and their formation is driven by weak multivalent interactions. There are difficulties with fixing condensates both in cells and in vitro (antibody penetration can be a major problem, amongst other issues). It is likely beyond scope to repeat all experiments with live imaging. But, as this study uses fluorescent proteins, could the authors perform representative live-imaging-based examples that demonstrate that the fixation is not compromising their main points? Also, a discussion on fixation and how this plays a role in disrupting condensate systems would be helpful and informative.

We thank the reviewer for highlighting this important point and agree that fixation is a well-known and important issue when considering condensation artifacts. To emphasize this, we expanded our text and conducted additional live-cell experiments using cell lines expressing DDX3X and EDC3 tagged with key fluorescent proteins (EGFP, mEGFP, EYFP, and mCherry) (new panels: EV Fig 1C/D and EV Fig 3 C/D). These live-cell results show trends that closely mirror those observed in fixed cells.

We added the following panels and statements to the manuscript:

“Furthermore, to rule out fixation-induced artifacts³ that might differentially affect fluorescent proteins and thereby alter MLO appearance, we conducted the experiments in both fixed and in living cells (EV Fig. 1C/D). The results were largely consistent with those observed in transient transfections, especially for DDX3X-mCherry which still showed a drastic reduction of SG numbers.”

EV Fig 1C/D:

(C/D) Stable inducible HeLa cell lines were induced with doxycycline for 24 h to express the respective DDX3X-tag constructs. Cells were stressed with 500 µM sodium arsenite for 30 min and imaged live. Scale bar: 20 µm. Quantification of the number of DDX3X foci and their area (D). N = 3, n ≥ 110 cells.

“Again, similar results were obtained in both in fixed and in living cell lines (EV Fig. 3C/D). This suggests that EDC3 tagging affects not only its own targeting to PBs but also the recruitment of other proteins such as DCP1 or DDX6.”

EV Fig. 3 C/D:

(C/D) Stable inducible HeLa cell lines were induced with doxycycline for 24 h to express the respective EDC3-tag constructs. Cells were stressed with 500 µM sodium arsenite for 30 min and imaged live. Scale bar: 20 µm. Quantification of number of EDC3 foci and their area (D). N = 3, n ≥ 100 cells.

Additionally, we would like to point out that all yeast experiments in this study were performed in living cells, in which we also observe pronounced tag-dependent effects. To make this clearer we added the following phrase:

“Using live cell imaging, we observed striking differences in SG numbers depending on the FP used:...”

Can you include a reference where it is shown that DCP1 is a good marker for PBs? What is your metric for a good marker? Perfectly overlapping to diffraction limit?

This is a valid point, and if you have a specific opinion we would be interested to hear it. We picked DCP1 as it has been used as a PB marker in recent prominent publications on PB (e.g., PMID 38548718, 39577428). Furthermore, to validate it as a marker protein for our

experiment, we performed colocalization analysis with EDC3 and observed complete overlap (Appendix Fig. 4A). We have now also included immunofluorescence against another PB marker protein, DDX6 (Appendix Fig. 4C/D and quantification also in EV Fig. 3E).

Appendix Figure 4:

(A) Stressed HeLa K cells (500 μ M sodium arsenite, 30 min) were fixed with methanol and immunostained for EDC3 and DCP1 with respective antibodies. Scale bar: 10 μ m. Line plot shows the normalized intensity of EDC3 and DCP1 across a P-body.

(B) Tag-EDC3 expression levels of stable inducible HeLa cell lines from Figure 3C were analyzed by immunoblotting with the indicated antibodies.

(C/D) Stable inducible HeLa cell lines were induced with doxycycline for 24 h to express the respective EDC3-tag constructs. Cells were stressed with 500 μ M sodium arsenite for 30 min before fixation. Cells from the same well were either immunostained for DDX6 or DCP1 with respective antibodies. Scale bar: 20 μ m. Quantification of number of DDX6 and DCP1 foci (I). N = 3, n \geq 140 cells.

EV Fig 3:

(E) Stable inducible HeLa cell lines were induced with doxycycline for 24 h to express the respective EDC3-tag constructs. Cells were stressed with 500 μ M sodium arsenite for 30 min before fixation. Cells from the same well were either immunostained for DDX6 or DCP1 with respective antibodies. Scale bar: 20 μ m. Quantification of number of DDX6 and DCP1 foci (I). N = 3, n \geq 140 cells.

Can you provide an explanation and include in text why the mCherry-tagged and the G3BP1 localization is different?

Thank you for this comment, this is indeed an important and interesting point. We think this reflects impaired recruitment of mCherry-tagged DDX3X to stress granules, a phenomenon less pronounced with other fluorescent protein tags. To highlight this observation, we expanded the text of the results section. Following the comment of another reviewer, we also added a co-staining with a DDX3X antibody for cells expressing DDX3X-FP to also visualize endogenous (+exogenous) DDX3X and observed the same effect (EV Fig. 1E/F and Appendix Fig. 3F)

We changed the text to now read as follows, and added the following panels:

“To test whether endogenous DDX3X is affected by exogenous expression of FP-tagged DDX3X, we stained the stable cell lines with a DDX3X antibody. The resulting immunofluorescence signal was very similar between parental cells and those expressing DDX3X-FP (EV Fig. 1 E/F), suggesting that overall endogenous DDX3X levels and localization are not substantially altered. We then immunostained for another SG marker, G3BP1, which also showed minimal differences in SG appearance between parental and DDX3X-FP cells (Fig. 1E/F, Appendix Fig. 3F). In both co-stainings, we observed a marked reduction in SG-enrichment for DDX3X-mCherry compared to the antibody signal (Appendix Fig. 3D/F). This suggests that while recruitment of DDX3X-mCherry is reduced, localization of endogenous SG proteins, such as G3BP1 and endogenous DDX3X, remains unaffected.”

EV Figure 1:

(E/F) Stable inducible HeLa cell lines were induced with doxycycline for 24 h to express the respective DDX3X-tag constructs. Cells were stressed with 500 μ M sodium arsenite for 30 min. Cells were also stained for DDX3X with immunostaining. Scale bar: 20 μ m. Quantification of the number of FP-DDX3X and DDX3X foci detected by the DDX3X antibody (F). N=3. n \geq 40 cells.

Appendix Fig 3:

(F) Stable inducible HeLa cell lines were induced with doxycycline for 24 h to express the respective DDX3X-tag constructs. Cells were stressed with 500 μ M sodium arsenite for 30 min. Cells were also stained for DDX3X with immunostaining. Additional quantifications for images displayed in EV Fig 1E. N=3. n \geq 40 cells.

Can you give more background on the Calvados system when it is introduced?

Thank you for this suggestion. We added additional text introducing the calvados model, and hope it helps with understanding.

“Building on previous studies establishing a molecular grammar for phase separation³⁷ and molecular simulation methods that relate protein sequence to phase separation propensities³⁸, we performed direct-coexistence simulations on phase-separating proteins or intrinsically disordered regions (IDRs) fused to various FPs using the CALVADOS (Coarse-graining Approach to Liquid–liquid phase separation Via an Automated Data-driven Optimization Scheme) 3 coarse-grained model³⁸. CALVADOS 3 is a coarse-grained (one bead per residue) molecular dynamics simulation model that generates conformational ensembles of intrinsically disordered proteins (IDPs) and multi-domain proteins, from which the phase behavior of these

proteins can be inferred. It incorporates a harmonic potential for bonded interactions, an Ashbaugh-Hatch (AH) potential to model van der Waals and hydrophobic forces, a Debye-Hückel (DH) potential for salt-screened electrostatic interactions, and an additional harmonic potential to maintain the structural integrity of folded domains within multi-domain proteins³⁸.

Referee #3:

In the manuscript "Fluorescent protein and peptide tags alter condensate formation and dynamics in vivo and in vitro" the authors test the effect of a range of fluorescent tags on DDX3X, Pab1, EDC3, CsdA, FUS and Nab2 condensate forming proteins. This work is a thorough effort to map the effect of commonly used protein tags on a number of often studied condensate forming proteins. While effects are often quite target-protein specific, the authors do provide actionable insights and provide useful recommendations for experimental design. The study is well designed, executed and represented. A few other studies have recently addressed the same tagging concern (Barkley et al., 2024; Pandey et al., 2024; Fatti, Khawaja and Weis, 2025), but all focus on a single or small selection of proteins of interest. Therefore, this work represents the first systematic effort to catalogue and understand tag-effects on biomolecular condensates and will provide an important resource for the condensate biology field.

I have only a few minor comments and suggestions for improvement that the authors may wish to address.

We thank the reviewer for their positive and encouraging feedback. We are particularly appreciative of their recognition of the systematic nature of our study, which sets it apart from other recent publications, as well as of the value of the practical guidelines we provide.

Fig 1A. The effects of the different tags are tested using transient over-expression, which leads to highly variable expression levels cell to cell, all the way from no expression to several-fold over endogenous levels. This is evidenced by the variability displayed in the different quantification methods. How reliable is the interpretation of the different tag-effects under such high variability? A more controlled experiment would be endogenous labeling of DDX3X with the different labels, but I appreciated this is a significant effort. At least the authors should compare the variability of the transiently expressed, tagged versions with that of the IF detected endogenous DDX3X, to evaluate the level of variability that is introduced by transient over-expression.

We thank the reviewer for this comment, it brings up a very interesting point. Since we were aware of the big variations in expression levels in transient transfections (e.g. see Figure 1 A/B, EV Fig 1A/B for DDX3X and Fig. 3A/B and EV Fig. 3A/B for EDC3), we had already in the initial submission performed experiments in stable cell lines for all key proteins of interest expressing the tagged POI at 1:1 level with the endogenous protein based on Western blotting.

In the previous version these figures for DDX3X were in the supplement and only their quantification was displayed in the main figure. Considering the importance of this experiment, we now show the images of stable cell lines for DDX3X more prominently in the main Figure 1E/F. In addition, we have now quantified the variability of expression in these cell lines, and see very little variation of expression between individual cells (see Appendix Fig. 3D/E). Additionally, we created cell lines stably expressing FP-EDC3, which are also very homogenous, see Fig. 3C/D.

Appendix Fig 3:

(B/C) Stable inducible HeLa cell lines were induced with doxycycline for 24 h to express the respective DDX3X-tag constructs at endogenous level. Cells were stressed with 500 μ M sodium arsenite for 30 min before PFA fixation. Quantification of DDX3X foci as displayed in Figure 1E: number (B) and area (C) relative to DDX3X-tag expression level. $N = 3$, $n \geq 65$ cells.

The authors find an interesting lack of correlation between the level of over-expression and the propensity to form condensates that flies in the face of LLPS mechanics but is more widely observed in the condensate field. Can the authors comment on why there is no strong correlation between the level of protein and condensate formation?

Thank you for bringing up this point, it is something we have wondered about as well. For SRRM2, we observe a clear correlation between expression level and condensate area, consistent with what is seen for single-component systems *in vitro*, and supporting the idea that SRRM2 expression levels are a major driver of nuclear speckle appearance (see Appendix Figure 5).

For the other proteins we express, we suspect that they are recruited to pre-existing condensates. In the context of transient overexpression, even the lowest expression levels may already saturate available binding sites, and other factors become limiting. Additionally, differences in the material properties of the condensates may play a role — SRRM2-containing condensates might be more dynamic and ‘flexible’ for incorporation of additional protein compared to others.

To reflect this, we have expanded the following statement:

“Surprisingly, for most tags, we observed only a weak correlation between expression levels and SG number or area, which could suggest that tagged DDX3X is primarily recruited to pre-existing membraneless organelles (MLOs), rather than inducing de novo condensate formation as a result of overexpression.”

Do the exogenously expressed tagged proteins also affect endogenous condensates? Can the authors perform an IF for DDX3X together with the labeled proteins and see if 1) the exogenous protein partitions into endogenous condensates and 2) if the effects of the specific tags also apply to endogenous DDX3X condensates? Currently only G3BP1 is used as a marker.

Thank you for this comment and suggesting this important experiment, which we have performed. In agreement with our G3BP1 co-staining data, anti-DDX3X immunofluorescence of cells expressing exogenous DDX3X-FP shows that all observed foci co-localize. This

indicates that the exogenous fusion protein and endogenous DDX3X localize to the same structures.

Importantly, the antibody signal indicates that recruitment of the endogenous / antibody-stained DDX3X protein is not affected by exogenous expression of tagged DDX3X-FP. This is particularly visible for cells expressing DDX3X-mCherry which does only partially partition into SGs: in these cells, the DDX3X IF signal is comparable to the parental cell line.

We have added this experiment as panel EV Fig. 1E/F, Appendix Fig. 3F to the manuscript and changed the text as follows.

EV Figure 1:

(E/F) Stable inducible HeLa cell lines were induced with doxycycline for 24 h to express the respective DDX3X-tag constructs. Cells were stressed with 500 μ M sodium arsenite for 30 min. Cells were also stained for DDX3X with immunostaining. Scale bar: 20 μ m. Quantification of the number of FP-DDX3X and DDX3X foci detected by the DDX3X antibody (F). N=3. n \geq 40 cells.

Appendix Fig 3:

(F) Stable inducible HeLa cell lines were induced with doxycycline for 24 h to express the respective DDX3X-tag constructs. Cells were stressed with 500 μ M sodium arsenite for 30 min. Cells were also stained for DDX3X with immunostaining. Additional quantifications for images displayed in EV Fig 1E. N=3. n \geq 40 cells.

“To test whether endogenous DDX3X is affected by exogenous expression of FP-tagged DDX3X, we stained the stable cell lines with a DDX3X antibody. The resulting immunofluorescence signal was very similar between parental cells and those expressing DDX3X-FP (EV Fig. 1 E/F), suggesting that overall endogenous DDX3X levels and localization are not substantially altered. We then immunostained for another SG marker, G3BP1, which

also showed minimal differences in SG appearance between parental and DDX3X-FP cells (Fig. 1E/F, Appendix Fig. 3F). In both co-stainings, we observed a marked reduction in SG-enrichment for DDX3X-mCherry compared to the antibody signal (Appendix Fig. 3D/F). This suggests that while recruitment of DDX3X-mCherry is reduced, localization of endogenous SG proteins, such as G3BP1 and endogenous DDX3X, remains unaffected.”

Fig 3C/D shows that tag-EDC3 affects DCP1 in P-bodies. Is the condensate-perturbing effect also POI dependent?

Thank you for this insightful question. We have performed additional experiments using DDX6, which is a core PB protein and also frequently used as a PB marker. We find that expression of FP-EDC3 leads to a loss of the anti-DDX6 immunofluorescence signal, similar to what we observed with anti-DCP1 (new panels in Appendix Fig. 4 C/D, EV Fig 3E). In contrast, when we express C-terminally tagged DDX6 (mCherry, mEGFP, or V5), P-bodies remain intact (new panels Fig EV 3F/G). This further supports the conclusion that the impact of tagging is highly dependent on the specific POI being tagged.

We added the following panels and changed the result section of the text as follows.

Appendix Figure 4:

(C/D) Stable inducible HeLa cell lines were induced with doxycycline for 24 h to express the respective EDC3-tag constructs. Cells were stressed with 500 μM sodium arsenite for 30 min before fixation. Cells from the same well were either immunostained for DDX6 or DCP1 with respective antibodies. Scale bar: 20 μm. Quantification of number of DDX6 and DCP1 foci (I). N = 3, n ≥ 140 cells.

EV Fig 3:

(E) Stable inducible HeLa cell lines were induced with doxycycline for 24 h to express the respective EDC3-tag constructs. Cells were stressed with 500 μ M sodium arsenite for 30 min before fixation. Cells from the same well were either immunostained for DDX6 or DCP1 with respective antibodies. Scale bar: 20 μ m. Quantification of number of DDX6 and DCP1 foci (I). N = 3, n \geq 140 cells.

EV Fig 3:

(F/G) HeLa K cells were transiently transfected with DDX6 plasmids for 24 h and stressed with 500 μ M sodium arsenite for 30 min. Cells from the same well were either immunostained for DDX6 or DCP1 with respective antibodies. Scale bar: 20 μ m. Quantification of number of DDX6 and DCP1 foci (G). N = 3, n \geq 85.

“Stable cell lines expressing tagged EDC3 at endogenous levels confirmed that mCherry considerably reduced PB foci numbers (Fig. 3C/D, Appendix Fig. 4C/D), and surprisingly demonstrated that mCherry-EDC3 expression also drastically reduced the number of co-stained DCP1 or DDX6 foci, while other FPs led to a slight increase (Fig. 3D, EV Fig. 3E, Appendix Fig. 4C/D).”

“Similarly, C-terminal tagging of DDX6 did not alter PB appearance (EV Fig. 3F/G).”

While it's important to know the effect of tags on condensate formation, a typical experimental setup compares two variants of the same protein (e.g. wt v.s. disease mutant) both tagged with the same fluorophore. In this case the tag-effects are internally controlled since both proteins carry the same tag. Can the authors comment on or experimentally address the risk of tags unduly affordable ng such a comparison?

Thank you for this suggestion, we agree that this is an important point. We added the following sentences to the discussion:

“These considerations are also relevant when comparing wild-type and mutant proteins. Since point mutations typically do not substantially alter surface properties or condensation behavior, they are generally expected to be influenced by tagging in a similar way as the wild-type POI. Truncation mutants often exhibit more pronounced changes in molecular and phase-separation properties, necessitating careful evaluation of both tag choice and position.”

Fig 4D. The authors claim HisTS alters RNA specificity but show no evidence for that. Instead an increased condensate propensity across the board is observed.

We thank the reviewer for this suggestion. We agree it is not fully accurate and have removed the mention of ‘specificity’ from the text.

References

Barkley, R.J.R. et al. (2024) "Fluorescent protein tags affect the condensation properties of a phase-separating viral protein.," *Molecular Biology of the Cell*, 35(7), p. ar100. doi:10.1091/mbc.E24-01-0013.

Fatti, E., Khawaja, S. and Weis, K. (2025) "The dark side of fluorescent protein tagging-the impact of protein tags on biomolecular condensation.," *Molecular Biology of the Cell*, 36(3), p. br10. doi:10.1091/mbc.E24-11-0521.

Pandey, N.K. et al. (2024) "Fluorescent protein tagging promotes phase separation and alters the aggregation pathway of huntingtin exon-1.," *The Journal of Biological Chemistry*, 300(1), p. 105585. doi:10.1016/j.jbc.2023.105585.

Dear Maria,

Thank you for the submission of your revised manuscript to EMBO reports. I have already forwarded you the referee reports, which are very positive with all three referees supporting publication after a couple of minor text edits.

Browsing through the manuscript myself, I noticed a few editorial things that we need before we can proceed with the official acceptance of your study.

- 1) Please rename the Data and resource availability section to "Data Availability". Please remove the sentence "Further information and requests for resources and reagents should be directed to the corresponding author, Maria Hondele (maria.hondele@unibas.ch)." as this section should exclusively refer to datasets deposited in external repositories.
- 2) Regarding the Author Contributions, we now use CRediT to specify the contributions of each author in the journal submission system. Therefore, please remove the Author Contributions from the manuscript file and make sure that the author contributions in our online manuscript tracking system are correct and up-to-date. The information you specified in the system will be automatically retrieved and typeset into the article. You can enter additional information in the free text box provided, if you wish.
- 3) Please update the references to the alphabetical Harvard style. The abbreviation 'et al' should be used if there are more than 10 authors. DOIs should only be used for preprints and datasets that have not been published yet. If you cite preprints, please add the prefix "preprint:" in the main text and the tag [PREPRINT] at the end of the reference in the reference list. In text citation is e.g., (preprint: Zhou Z et al, 2023).
- 4) The following funding information needs to be entered in the online manuscript tracking system: "Fellowships for Excellence" from the International PhD Program in Molecular Life Sciences of the Biozentrum, University of Basel.
- 5) The correct nomenclature for EV figures is Figure EV1 etc instead of EV Figure 1 or Expanded View Figure 1. Please correct callouts, figure legends, and the titles of the EV figures in our online manuscript tracking system accordingly.
- 6) The correct nomenclature for Appendix Figures and Tables is: Appendix Figure S1 etc and Appendix Table S1. Please correct the callouts, figure labels and legends accordingly.
- 7) Supplemental Table S1 is an incorrect callout.
- 8) The Appendix table of content needs page numbers.
- 9) You provided the Appendix Tables separately as an Excel file with 10 sheets. Each of these tables should either be moved to the Appendix file below its legend (Appendix Table S1-S10), or uploaded separately as EV tables (Table EV1-EV10, the legends would then be removed from the Appendix file and added as a separate tab within the respective .xls file).
 - Table S4, S6, S7, and S10 seem to be redundant with the information in the Reagents and Tools table and should be removed.
 - Table S1 is certainly best uploaded as Table EV1, to keep it in .xls format.
- 10) Please remove the Reagents and Tools table from the manuscript. The uploaded separate file is sufficient and will be typeset into the article by our production team. That said, please remove the "Instructions" statement from the file.
- 11) Our production/data editors have asked you to clarify several points in the figure legends (see below). Please incorporate these changes in the manuscript and return the revised file with tracked changes with your final manuscript submission.
 - Please provide the exact p values in the legend of figure EV4 B (unless it is $p < 0.0001$).
 - Please note that the box plots need to be defined in terms of minima, maxima, centre, bounds of box and whiskers, and percentile in the legends of figures 1B, D, F; 2G, 3B, D, F, H; 5B, EV1 D, F; EV2 H, EV3 A, D, E, G
 - Please note that the error bars are not defined in the legend of figure 5E
 - Please note that the scale bar needs to be defined for figure 1A
 - Please add a scale bar for Figure EV6B.
 - Scale bar size should only be defined in the legends, not with numbers in the figure panel/images.
- 12) We perform a routine image integrity check on all revised manuscript. Doing so, the following came to our attention:
 - The DDX3X-mEGFP image from Figure EV2C (200 mM) is shown again in Fig. EV5B (here as 'high pH'). This reuse must be clearly stated in the respective figure legends and justified. If appropriate, using another representative image is preferred.
 - The CsdA + ATTO488 image from Figure EV5B is reused in Appendix Figure S6B. Again, please clearly state this reuse in the figure legends and please justify it.
 - The SRRM2-mEGFP images in Figure 3G are shown again in Appendix Figure S5A as co-staining with SON. These re-use must clearly be stated in the respective figure legends (and justified).
 - In Appendix Figure S4E and F the EGFP-G3BP1 quantifications look very similar to each other.

- Our software further detected potential aberrations in the following images, which we would like to re-evaluate based on the source data:

- Figure 1E
- Figure 2A
- Figure EV3F
- Appendix Figure 4C

Could you please send us the unmodified source data for these figure panels? I know that all imaging data have been uploaded to BiImage Archive, but having these data specifically at hand, would speed up our checks. Thank you.

- Please describe your findings in the abstract in present tense.

- Finally, EMBO Reports papers are accompanied online by

A) a short (1-2 sentences) summary of the findings and their significance,

B) 2-3 bullet points highlighting key results and

C) a schematic summary figure that provides a sketch of the major findings (not a data image).

Please provide the summary figure as a separate file in PNG or JPG format at a size of 550x300-600 pixels (width x height).

Please note that the size is rather small and that text needs to be readable at the final size. Please send us this information along with the revised manuscript.

With kind regards,

Martina

=====

Referee #1:

I thank the authors for addressing all of my concerns and addressing the other reviewer concerns. The addition of endogenous Ab staining (in some cases), additional live-imaging experiments, and improved computational analysis together strengthens this already-important manuscript for the condensate field (but also generally applicable to the bioimaging field).

Only two minor questions:

(1) For the computationally-determined DDX4 csat values, what is the error in measuring these?

(2) In the introduction, while mentioning other examples about tag effects on condensates, could include the following reference:
DOI: 10.1016/j.bpj.2023.11.3401

Referee #2:

The authors have adequately addressed my comments, concerns, and suggestions to their manuscript. It is an exciting and timely piece of work and should be published in EMBO Reports.

Referee #3:

The authors have adequately addressed all my concerns and, together with the edits based on the other reviewers, have elevated the quality of their manuscript even further.

Great paper that should be heeded by all in our field!

(1) For the computationally-determined DDX4 c_{sat} values, what is the error in measuring these?

We have added the following statement to the methods section:

Errors of the saturation concentrations reported in Table S3 correspond to standard errors estimated using the “BLOCKING” method, in which trajectories were divided into optimally sized blocks to assess uncertainty (<https://github.com/FrPsc/BLOCKING>).

(2) In the introduction, while mentioning other examples about tag effects on condensates, could include the following reference: DOI: 10.1016/j.bpj.2023.11.3401

We have added this reference to the text twice (see point 3 below).

The authors have addressed all editorial requests.

Dr. Maria Hondele
University of Basel
Biozentrum
Spitalstrasse 41
Basel, Basel 4056
Switzerland

Dear Maria,

I am very pleased to accept your manuscript for publication in the next available issue of EMBO reports. Thank you for your contribution to our journal.

Kind regards,

Martina
